# Time-Masked Transformers with Lightweight Test-Time Adaptation for Neural Speech Decoding

**Ebrahim Feghhi**[1,2*]  **Shreyas Kaasyap**[2]  **Nima Hadidi**[1,2]  **Jonathan C. Kao**[1,2,3]

[1] Neuroscience Interdepartmental Program
[2] Department of Electrical & Computer Engineering
[3] Department of Computer Science
University of California, Los Angeles

## Abstract

Speech neuroprostheses aim to restore communication for people with severe paralysis by decoding speech directly from neural activity. To accelerate algorithmic progress, a recent benchmark released intracranial recordings from a paralyzed participant attempting to speak, along with a baseline decoding algorithm. Prior work on the benchmark showed impressive accuracy gains. However, these gains increased computational costs and were not demonstrated in a real-time decoding setting. Here, we make three contributions that pave the way towards accurate, efficient, and real-time neural speech decoding. First, we incorporate large amounts of time-masking during training. On average, over $50\%$ of each trial is masked. Second, we replace the gated recurrent unit (GRU) architecture used in the baseline algorithm with a compact Transformer. The Transformer architecture uses $83\%$ fewer parameters, cuts peak GPU memory usage by $52\%$, and is significantly faster to calibrate relative to the GRU. Third, we design a lightweight variant of an existing test-time adaptation method developed for decoding handwriting from neural activity. Our variant adapts the model using multiple time-masked augmentations of a single trial and requires only one gradient step per trial. Together, these contributions reduce word error rate by over $20\%$ and effectively mitigate performance degradations across held-out days in a real-time decoding setting while substantially lowering computational costs.

## 1 Introduction

Conditions including amyotrophic lateral sclerosis (ALS) and brainstem stroke can lead to severe paralysis, leaving individuals unable to speak or interact with the world. A promising path toward restoring communication for these individuals is speech neuroprostheses, which bypass the vocal apparatus and decode speech directly from neural activity (35). Speech neuroprostheses have made significant strides in recent years, demonstrating the ability to decode speech with high accuracy over large vocabularies (41; 30; 9; 25).

To be viable in real-world clinical settings, decoding algorithms for speech neuroprostheses should ideally satisfy several key criteria beyond accuracy. First, they be able to operate in a real-time "streaming" fashion, decoding speech with low-latency over short windows rather than waiting for the entire utterance to finish (25). Second, they should have low computational requirements to enable on-device inference and adaptation (8), minimizing reliance on external connections and preserving user privacy. Finally, decoding algorithms should be easily integrated with test-time adaptation methods that mitigate performance degradation across time (17).

---

[*]Correspondence: `ebrahimfeghhi@g.ucla.edu`

39th Conference on Neural Information Processing Systems (NeurIPS 2025).

To accelerate progress along these lines, the Brain-to-Text Benchmark '24 was released, an open-source dataset containing intracortical neural recordings while a participant with ALS attempted to speak sentences across 24 days. Along the with the dataset, the organizers provided a baseline decoding algorithm which consisted of a gated recurrent unit (GRU) architecture to decode neural activity into phonemes, followed by beam search guided by an n-gram language model (LM) [41].

A recent summary report outlined the findings of the top four entries submitted to the benchmark [42]. Several entries replaced the GRU architecture with Transformers, deep state space architectures, and convolutional neural networks (CNNs), and found that none of these architectures outperformed the baseline GRU. For example, the first place entry reported that the phoneme error rate (PER) for Transformers was more than double that of the baseline GRU [24]. While researchers found that using a learning rate scheduler, layer normalization [5], and different optimization algorithms helped improve accuracy, the modification that led to the largest improvement was "using an ensemble of neural decoders to generate a diverse set of sentence hypotheses, and then using a fine-tuned large language model (LLM) to merge these hypotheses into a finalized sentence" [42], an idea that was first implemented by [6] in the context of neural speech decoding.

Although the accuracy gains from prior work are remarkable, there are several important caveats. First, the top four entries all used a bidirectional GRU, which requires access to future neural activity and therefore cannot decode speech in real-time. Furthermore, LLM merging was only applied after the entire text was decoded, as applying it to intermediate outputs is not straightforward. Second, some entries used up to 10 bidirectional GRUs and GPT 3.5, which makes inference and test-time adaptation on local resource-constrained devices challenging. Third, since the sentences used in the benchmark were from the publicly available Switchboard corpus [2], it is possible LLMs were trained on this corpus and so their contribution is overstated relative to conversational settings. These points highlight how optimizing for a single metric on machine learning benchmarks can lead to sacrifices along other dimensions important for real-world use.

In this work, we aimed to holistically improve speech neuroprostheses by focusing on multiple criteria: accuracy, capability for real-time streaming, low computational costs, and robustness to distribution shifts. In order to do so, we focused on improving the neural network that translates neural activity into phonemes rather than external language modeling components. Our improvements are based on two key observations. First, the baseline GRU overfits early in training (Figure 1a). Second, the GRU processes highly redundant inputs at each time step in order to achieve optimal performance: consecutive inputs are 87.5% overlapping when using the settings proposed by Willett et al. [41] (Figure 1b), which we hypothesized is a source of inefficiency.

Based on these observations, we proposed two modifications to the baseline algorithm. First, in order to delay overfitting, we incorporated time-masking which is a regularization strategy that masks contiguous temporal chunks of the input neural activity during training (Figure 1c) [32]. Second, we replaced the GRU architecture with a compact, unidirectional Transformer (Figure 1c) [38]. We hypothesized that the GRU requires overlapping inputs due to its lossy memory, which may result in increased computational costs. Since Transformers have perfect memory for a fixed context length, they are likely able to efficiently processes non-overlapping inputs. To foreshadow the results, the Transformer trained with time-masking (time-masked Transformer) achieved a WER of 12.17% with a 3-gram LM and 8.18% with the 5-gram LM setup, which is 20% and 26% lower than the baseline unidirectional GRU, respectively. The Transformer architecture also substantially reduces parameter count, peak GPU memory usage, FLOPs, and per-epoch training times.

We next leveraged the time-masked Transformers to improve upon the test-time adaptation method created by Fan et al. [17] for handwriting neuroprostheses. Their approach, **C**ontinual **O**nline **R**ecalibration with **P**seudo-labels (CORP), refines the text produced by a GRU-based model with an n-gram LM, and then adapts the GRU to output the LM-refined text during test-time. CORP utilizes multiple gradient steps per trial and maintains previous data to enable adaptation. Here, we present "DietCORP", a lightweight variant which enables test-time adaptation in a single gradient step without storing previous data by leveraging multiple time-masked augmentations of the same trial (Figure 1d). When combined with the low memory requirements and fast training speeds of the Transformer, DietCORP effectively adapts the model to distribution shifts while requiring only 1.3 GiB of peak GPU memory usage and 18 ms to adapt per trial.

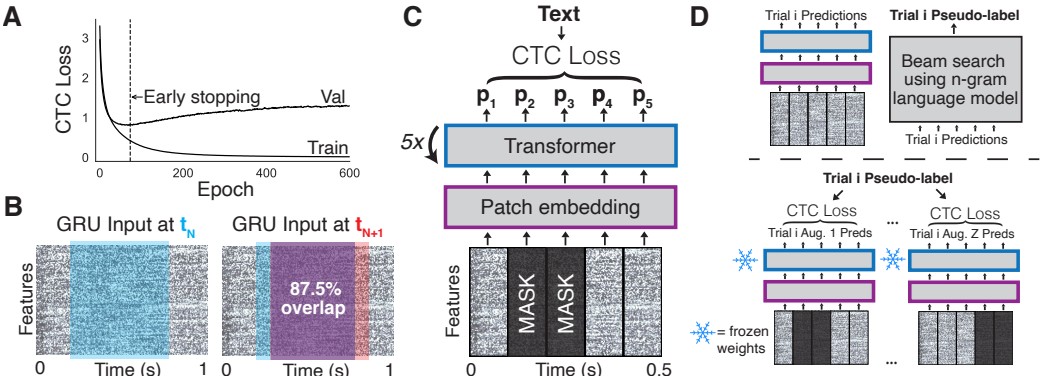

Figure 1: **A.** The GRU exhibits pronounced overfitting when training for long durations. Black dashed line indicates where training was stopped for the baseline model. **B.** Adjacent input windows to the GRU overlap by 87.5% when using the optimal baseline hyperparameters (window length = 640 ms, stride = 80 ms). **C.** We replaced the GRU with a lightweight Transformer-based model. The Transformer takes as input non-overlapping temporal patches of neural activity and outputs logits (denoted as $\mathbf{p}_i$). Consecutive patches were replaced with a MASK token during training, as denoted by dark coloring. We used the connectionist temporal classification (CTC) loss. **D.** An overview of DietCORP. In the top panel, the Transformer architecture is run in evaluation mode to generate logits, and these logits are integrated with a language model guided beam search to generate a pseudo-label. In the bottom panel, the model is trained to produce the pseudo-label across $Z$ time-masked augmentations with CTC loss. Only the patch embedding module is adapted during this process.

## 2  Related Work

Although we found success with using a Transformer, Transformers are typically not the architecture of choice for speech neuroprostheses or brain computer interfaces (BCIs) more broadly. Willsey et al. ([43]) used a convolutional neural network (CNN) to decode finger movements. Costello et al. ([12]) followed up on this work, and showed that recurrent neural networks (RNNs) achieve better finger movement decoding performance than either CNNs or Transformers. The majority of speech neuroprostheses have employed some combination of CNNs and RNNs ([3; 37; 28; 40; 41; 30; 4; 9; 36; 25]), although a few studies have used Transformers ([39; 11; 10]). In the discussion of the Brain-to-Text benchmark results, the authors speculated that the relatively poor performance of Transformers may be attributed to the fact that **1)** more optimization is required or **2)** phonemes are represented over short context windows in neural activity, which suits GRUs because they naturally implement a locality bias whereas Transformers are better suited for modeling long-range dependencies ([42]).

Our Transformer architecture is most similar to that of Chen et al. ([10]) in the BCI literature. Specifically, we also used relative positional embeddings and temporal patches as input. However, each temporal patch in our work consists of data from all electrodes, whereas Chen et al. ([10]) assign each electrode to its own temporal patch. Assigning each electrode to its own patch can substantially increase the context size, and Chen et al. ([10]) use the Swin Transformer ([26]) to address this challenge. Beyond model architecture, Chen et al. ([10]) displayed their results on a closed-source dataset consisting of electrocorticographic (ECoG) and depth electrodes (sEEG), and they translated neural activity directly into speech. Our results are displayed using an open-source dataset consisting of microelectrode array (MEA) recordings, and we translated neural activity into text. Li et al. ([24]) also used Transformers on the Brain-to-Text Benchmark '24; however, to the best of our knowledge the authors did not release sufficient details for reproducing their Transformer results.

Structured input masking was initially popularized by Park et al. ([32]) for automatic speech recognition (ASR). Within speech neuroprostheses, Littlejohn et al. ([25]) masked entire channels and cropped each trial by applying a temporal window. Metzger et al. ([30, 29]) applied a single temporal mask to each trial. The masking augmentations used by Littlejohn et al. ([25]); Metzger et al. ([30, 29]) were primarily applied on close-sourced, ECoG datasets and mask a much smaller percentage of the input

than in this work. Furthermore, the contribution of masking relative to other augmentations is not emphasized in these works. Beyond speech neuroprostheses, Saeed et al. (34); Ding et al. (15) applied structured input masking for EEG-based BCIs, and Fu et al. (19) applied it for intracranial monkey BCIs.

DietCORP is most directly inspired by CORP (17). However, the idea to use multiple augmentations, rather than previous data, for test-time adaptation was inspired by Zhang et al. (46) and Yao et al. (44). Zhang et al. (46) encouraged models to make consistent and confident (low entropy) predictions across several augmentations of the same image at test-time. Yao et al. (44) add a KL divergence loss term across two time-masked versions of the same audio sample to the connectionist temporal classification (CTC) (21) loss during training.

## 3  Methods

### 3.1  Neural Dataset

The Brain-to-Text '24 benchmark dataset consists of microelectrode array (MEA) recordings from the ventral premotor cortex (area $6v$) of a single participant with ALS. Data was recorded from two microelectrode arrays with $64$ channels each, a ventral 6v array and a dorsal 6v array. Spike band power and threshold crossings were extracted for each channel, leading to a total of $256$ features. Neural activity was recorded while the patient attempted to speak 10,850 sentences. In each trial, the subject was shown a sentence, and attempted to speak the sentence at the onset of the "go" cue. All analyses were done on neural activity during the "go" phase while the participant attempted to speak the sentence. The neural activity was provided in $20$ ms time bins ($50$ Hz resolution), and z-scored within each block (20-50 sentences). For additional details, refer to (41).

### 3.2  Word error rate and phoneme error rate

We computed the *word error rate* (WER) as the Levenshtein (edit) distance between the predicted and target word sequences, normalized by the total number of words in the target. Formally, WER is defined as

$$WER = \frac{S + D + I}{N}$$

where $S$, $D$, and $I$ denote the number of substitutions, deletions, and insertions, respectively, and $N$ is the total number of words in the target transcription. Similarly, the *phoneme error rate* (PER) was computed in the same manner but at the level of phonemes rather than words. We applied an n-gram guided beam search process when computing WER (Section E), and computed PER using greedy decoding.

### 3.3  Train, validation, and test splits

The benchmark provided train, validation, and test splits. There were $8800$ sentences in train, $880$ sentences in validation, and $1200$ sentences in test. Train and validation sentences were recorded on $24$ days (collected over almost $4$ months), and test sentences were recorded on $15$ out of the $24$ days. We refer to the days with testing data as "competition" days. All hyperparameter tuning was performed on the validation set, and the validation set was not used for training after hyperparameter tuning for the main results (Section 4.1, 4.2). For DietCORP (Section 4.3), we were interested in evaluating its effectiveness on days with no training data. Since the original splits contained training data for all days, we modified the splits by holding out $5$ or $8$ competition days entirely from the training set. To evaluate performance, we used the original training and validation data from these held-out days as our new test set since the competition platform does not allow for evaluating WER on a subset of the competition days. We used all data from the preceding days for training.

### 3.4  Baseline gated recurrent unit-based model

The dataset was accompanied by a baseline gated recurrent unit (GRU)-based model. This model first passed the input features (spike band power and threshold crossings) through a day-specific linear layer followed by a softsign non-linearity, which served to account for day-specific differences in neural activity. At each step, the GRU received a vector of these day-transformed inputs with

dimension $F \cdot T_{in}$, where $F$ is the number of features (256) and $T_{in}$ (the window length) is the number of neural time bins. The model consisted of 5 GRU layers. The hidden state of the final layer was passed through a linear layer to produce logits over phonemes, the CTC blank token, and a silence token. Logits were output every $T_{out}$ (stride length) steps. For optimal performance, the baseline algorithm set $T_{in} = 32$ and $T_{out} = 4$, resulting in an $87.5\%$ overlap between consecutive inputs.

We followed the same training procedures as listed in the Pytorch codebase provided by Willett et al. (41) for the baseline GRU model. Specifically, we used the Adam optimizer (22) and connectionist temporal classification (CTC) loss (21). During training, white noise and baseline shift augmentations were added to the neural activity, followed by causal Gaussian smoothing. Training was performed for 10,000 batches ($\sim$ 73 epochs). The full set of hyperparameters is listed in Table 5. We used a unidirectional GRU for all results unless otherwise stated.

## 3.5 Transformer-based model

Inspired by Dosovitskiy et al. (16) and Chen et al. (10), we replaced the GRU architecture with a Transformer. We segmented the input neural data into non-overlapping temporal patches, each consisting of $T_{in}$ time bins and all (256) features. $T_{in}$ was set to 5, such that each patch captured 100 ms of neural activity. Each patch was then flattened into a $F \cdot T_{in}$ ($F$ is the number of neural features) dimensional vector and passed through a patch embedding module, consisting of the following layers in sequence: LayerNorm (5), Linear, and LayerNorm. The linear layer projected the patches from $F \cdot T_{in}$ to the transformer hidden dimension size. Time-masking (Section 3.6) and input dropout was applied to the output of the patch embedding module, followed by $L$ Transformer blocks ($L = 5$). Each Transformer block comprised a self-attention layer followed by a feed-forward network. We applied LayerNorm before the self-attention layer, and the feed-forward network included the following layers in sequence: LayerNorm, Linear, GeLU activation, Dropout, Linear, and Dropout. Residual connections were added around both the self-attention layer and the feed-forward network.

We used relative positional embeddings similar to the T5 architecture (33). Specifically, we added a learned bias to the attention scores based on the relative distance between two patches. A causal attention mask was also used to ensure that each patch could only attend to itself and previous patches. The attention operation was then defined as:

$$\text{Attention}(\mathbf{Q}, \mathbf{K}, \mathbf{V}) = \text{softmax}\left(\frac{\mathbf{Q}\mathbf{K}^\top}{\sqrt{d}} + \mathbf{B} + \mathbf{M}\right)\mathbf{V}$$

where $\mathbf{B}_{i,j} = \mathbf{b}(i - j)$, and $\mathbf{b} \in \mathbb{R}^{2L-1}$ was a learnable vector containing scalar bias values for each possible relative position. The index $(i - j)$ represents the relative distance between query position $i$ and key position $j$, allowing attention scores to incorporate relative position information up to a maximum absolute distance of $L - 1$. The matrix $\mathbf{M}$ was the causal attention mask, defined as:

$$\mathbf{M}_{i,j} = \begin{cases} 0 & \text{if } j \leq i \\ -\infty & \text{otherwise} \end{cases}$$

The outputs of the causal Transformer were passed through a final LayerNorm followed by an output linear layer to obtain logits over phonemes, the CTC blank token, and silence token. Logits were output every $T_{out}$ time bins, and we set $T_{out} = 5$.

When training the Transformer, we applied a log-transformation to the neural data before z-score normalization and trained for 250 epochs. We also used a learning rate scheduler, where we decreased the learning rate by a factor of 10 after 150 epochs. Training was performed with the AdamW optimizer (27) and CTC loss, and we also applied white noise and baseline shift augmentations followed by causal gaussian smoothing. The full set of Transformer hyperparameters is listed in Table 6. The code used for the Transformer model is largely based on the following GitHub repository: vit-pytorch.

## 3.6 Time-masking

We performed time-masking by masking several contiguous temporal patches (Transformer) or time bins (GRU) of each trial. The rationale for applying contiguous masking in addition to input dropout is that input dropout does not mask contiguous input chunks, and is therefore likely a weaker form

---

**Algorithm 1:** TIMEMASK($\mathbf{x}, N, M$)

---

**Input:** Trial $\mathbf{x} \in \mathbb{R}^{L \times C}$ ( $L$ patches, $C$ features);
$N$ — number of masks;
$M$ — max-mask length as a fraction of trial length ($0 < M \leq 1$)
**Output:** Masked trial $\tilde{\mathbf{x}}$
$F \leftarrow \lfloor M \cdot L \rfloor$ ;                                    // maximum mask length
**for** $k$ ***in*** $1{:}N$ **do**
    $S \sim \mathcal{U}(0, L - F)$ ;                                // start index
    $D \sim \mathcal{U}(0, F)$ ;                                       // mask length
    $\mathbf{x}[S : S + D] \leftarrow$ LEARNABLE-MASK TOKEN
**return** $\tilde{\mathbf{x}}$

---

of regularization. A description of the time-masking algorithm is provided in Algorithm 1 for the Transformer. For the time-masked GRU, all details are identical except masking was applied at the time bin level, and masked bins were replaced with $0$ rather than a special MASK token. We do not enforce time-masks to be non-overlapping.

For all analyses, we set $N = 20$ (number of masks) and $M = 0.075$ (max mask length as a fraction of trial length). Under these settings $53\%$ of a trial is masked out on average (Section J). We report performance with other hyperparameters in Section I.

### 3.7 DietCORP

For each trial, we generated a "pseudo-label" by applying language model guided beam search (Section E) to the model predictions, as done in CORP (17). "Pseudo-label" generation is already performed for real-world use, and so it does not incur additional computational costs. In CORP, the pseudo-label is combined with labeled training data as well as previously generated pseudo-labeled data to update model weights with CTC loss until either the loss decreases below a threshold or 200 gradient steps are completed. DietCORP reduces computational costs and complexity by eliminating the need to store previous data and performing adaptation with one gradient step per trial. This is done by adapting the model across $Z$ augmentations of the same trial. Model weights are not reset across trials (i.e., calibration is continuous), as done in CORP.

When applying DietCORP, we only updated the patch embedding module because 1) neural distribution shifts across days are likely a form of input distribution shift (as opposed to feature or output distribution shifts) and selectively fine-tuning input layers is effective when dealing with input distribution shifts (23), and 2) test-time adaptation can rapidly degrade when adapting the entire model (23). Augmented trials were generated by creating $Z$ copies of a given trial, and applying white noise, baseline shift, and time-masking to each copy. Time-masking served as the main form of augmentation. We set $Z = 64$ since this was the batch size used for training, although DietCORP was robust to lower values of $Z$ as discussed in the results. All hyperparameters were set to the values used during training. The learning rate was set to the mean of the learning rate before and after the learning rate step decay was applied, which was $5e - 4$. We additionally clipped the gradient norm, as done in CORP, to $0.5$.

## 4 Results

### 4.1 Comparison with baseline unidirectional GRU

We first compared the causal Transformer trained with time-masking (Section 3.5, 3.6), or time-masked Transformer, with the baseline unidirectional GRU model, or baseline GRU (Section 3.4). When using the 3-gram language model (LM) (Section E), the time-masked Transformer achieved a word error rate (WER) that was 3.08 absolute percentage points better than the baseline GRU (or a $20.2\%$ relative improvement), which was a significant decrease ($p < 0.05$; one-sided independent t-test across 10 seeds) (Table 1). When using the stronger 5-gram LM setup, with an additional second-pass rescoring using an unpruned LM and large language model (LLM) (Section E), the time-masked Transformer performed 2.94 absolute percentage points better than the baseline GRU

Table 1: Performance comparison using different language modeling setups. The values represent word error rates (WER) as a percentage (mean $\pm$ SEM across seeds). An em dash (—) denotes settings that were not evaluated. Values for the Linderman Lab and diphone decoding models are provided by the original authors. Results are reported across 10 seeds except for the "Fine-tuned LLMs" setting ($N = 1$), diphone decoding with the 5-gram LM setup ($N = 5$), and Linderman Lab GRU ($N = 1$).

| Model | 3-gram LM Setup | 5-gram LM Setup | Fine-tuned LLM |
|---|---|---|---|
| Baseline Unidirectional GRU | $15.25 \pm 0.16$ | $11.12 \pm 0.13$ | — |
| Time-Masked Causal Transformer | $12.17 \pm 0.22$ | $8.18 \pm 0.22$ | 5.68 w/ Llama 3.1 8B |
| Linderman Lab Bidirectional GRU | — | 8.0 | — |
| Diphone Decoding with Bidirectional GRU | — | $8.39 \pm 0.22$ | 5.77 w/ GPT 3.5 6.85 w/ Llama 3.1 70B |

(or a 26.4% relative improvement) ($p < 0.05$; one-sided independent t-test across $N = 10$ seeds) (Table 1). This is to our knowledge the largest improvement in accuracy over the baseline GRU on this neural dataset when using a streaming compatible architecture.

Beyond gains in accuracy, the time-masked Transformer also substantially reduced computational costs relative to the baseline GRU. The Transformer used 83% fewer parameters, cut peak GPU memory usage by 52%, reduced mega floating pointing operations (mFLOPs) by 43% (Section C), while shortening per-epoch training times by 58% (Table 4.1). These efficiency gains make the Transformer architecture well-suited for on-device test-time adaptation (8) and motivate the experiments in Section 4.3. We also observed that beam search decoding times, measured after producing logits, were approximately 3 times faster when using the time-masked Transformer relative to the baseline model (Section H). Furthermore, the Transformer does not require day-specific parameters unlike the baseline GRU, which we speculate is due to the layer normalization layers in the patch embedding module.

## 4.2 Comparison against top-performing benchmark entries

We next compared the time-masked Transformer with two of the top-performing, entries in the Brain-to-Text Benchmark '24: the Linderman Lab bidirectional GRU (Section F) (42) and diphone decoding with a bidirectional GRU (24). Both these entries are not streaming compatible, and we used the WER values provided by the original authors. When using the 5-gram LM, there was no significant difference in performance between either the time-masked Transformer ($N = 10$ seeds) and the Linderman Lab GRU ($N = 1$ seed) ($p > 0.05$; one-sample t-test) or the time-masked Transformer and the diphone decoding approach ($N = 5$ seeds) ($p > 0.05$; independent t-test).

We next evaluated the time-masked Transformer with the "Fine-tuned LLM" setup (Section G), which involved fine-tuning an LLM to correct the outputs of an ensemble of models. We fine-tuned Llama 3.1 8B (20) on the outputs of 10 seeds of the causal time-masked Transformer. The Transformer + Llama 3.1 8B combination performed on-par with the best performing entry in Willett et al. (42), specifically the diphone decoding GRU + GPT 3.5 (7) or Llama 3.1 70B (Table 1).

Beyond accuracy, the Transformer architecture provides large gains in computational efficiency over the bidirectional GRU (Table 4.1), and we used a smaller, open-source LLM (Llama 3.1 8B) relative to previous entries that used either GPT 3.5, which is not open-source, or Llama 3.1 70B (6; 24).

Table 2: Values are reported using a Nvidia GeForce RTX 3090 GPU. Epoch time is reported as mean $\pm$ standard deviation ($N = 1200$). *MFLOPS* stands for mega floating point operations per second.

| Model | Parameters (M) | Peak memory (GiB) | MFLOPS | Epoch time (s) |
|---|---|---|---|---|
| Bidirectional GRU | 135.4 | 11.21 | 1563.29 | $55.63 \pm 0.69$ |
| Unidirectional GRU | 56.7 | 5.55 | 634.81 | $25.88 \pm 0.55$ |
| Causal Transformer | 9.4 | 2.66 | 364.2 | $10.81 \pm 0.17$ |

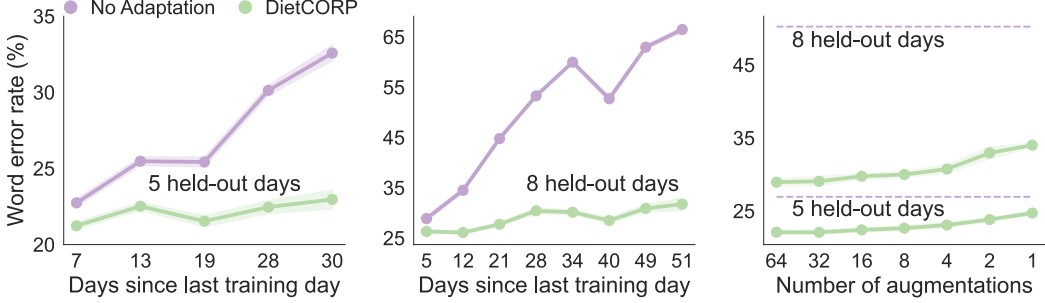

Figure 2: Results are for the time-masked Transformer with a 3-gram LM. Points show the mean over four seeds; shading indicates ±SEM. **A**. WER across five held-out days without adaptation and with DietCORP. **B**. Same as panel A with evaluation on eight held-out days. **C**. Average WER across held-out days as a function of the number of augmentations used by DietCORP. Green points are when using DietCORP; the purple dashed line is when no adaptation is performed. Lower curves correspond to five held-out days, upper curves to eight held-out days.

## 4.3 Test-time adaptation across held-out days

We next used the lightweight Transformer architecture to perform test-time adaptation (TTA) with DietCORP. In brief, DietCORP adapts the decoder to predict LM-refined pseudolabels (with the 3-gram LM) using multiple augmented versions of the current trial. DietCORP only requires one gradient step per trial and minimal hyperparameter tuning (Section 3.7). We applied DietCORP under two settings in which entire days were held-out from the training set: training on $15$ days and testing on the next $5$ competition days, and a more challenging setting where we trained on $12$ days and tested on the next $8$ competition days. In both cases, we compared performance on the held-out competition days when applying DietCORP versus when no TTA method was applied.

When training on $15$ days, there was a substantial rise in WER across the $5$ held-out competition days when no TTA method was applied, starting from a WER of $22.74 \pm 0.30\%$ (mean $\pm$ SEM across $N = 4$ seeds) on the first day and rising to a WER of $32.58 \pm 0.60\%$ by the fifth day. By contrast, when applying DietCORP the increase in WER was substantially smaller, beginning at $21.24 \pm 0.29$ and increasing to $22.97 \pm 0.72\%$ across the five days (Figure 2a). Under the more challenging evaluation setting, WER rose from $28.87 \pm 0.54\%$ to $66.47 \pm 0.67\%$ across the eight held-out days when no TTA was applied. DietCORP again substantially ameliorated the deterioration in performance, with WER ranging from $26.32 \pm 0.32\%$ to $31.74 \pm 1.41\%$ across held-out days (Figure 2b).

We next evaluated how DietCORP's benefit scales with the number of time-masked augmentations. Reducing the augmentation count progressively diminished its impact, supporting the idea that adaptation across multiple augmentations of the same trial helps buffer against test-time distribution shifts (46) (Figure 2c). Nevertheless, DietCORP still outperformed the no adaptation setting even when using only a single augmentation, indicating that there are still benefits when using a low number of augmentations. We finally quantified the memory requirements and adaptation times for DietCORP. When using the Transformer, DietCORP required 18 ms per trial and its peak memory

Table 3: Metrics were collected on an NVIDIA GeForce RTX 3090 over 2040 trials. *Peak memory* is the highest GPU memory recorded during DietCORP calibration across all trials. *Adaptation time* (reported as mean $\pm$ SD) is the time needed to update the model on a single trial with DietCORP. Only the patch-embedding layer was trainable for the Transformer; for a fair comparison, only the day-specific linear layer was trainable for the baseline GRU.

| Model | Peak memory (GiB) | Adaptation time (ms) |
|---|---|---|
| Time-masked Transformer | 1.33 | $18.21 \pm 5.34$ |
| Baseline GRU | 2.91 | $28.44 \pm 8.33$ |

usage was 1.3 GiB (Table 4.3). Applying DietCORP to the baseline GRU increased adaptation times by $56\%$ and peak memory usage by $118\%$, underscoring the benefit of the Transformer for on-device test-time adaptation.

## 4.4 Ablations to the time-masked Transformer

We conducted an ablation study to isolate the contributions of each component in our time-masked Transformer model (Table 4). All ablation results are reported on the validation set with a 3-gram LM. First, we removed two components from our training pipeline for the time-masked Transformer: the neural data log-transformation and the learning rate scheduler (Section 3.5). Removing either component independently increased the WER by $4\%$ relative to the original model. Next, we replaced the T5-style relative positional embeddings (Section 3.5) with absolute sinusoidal embeddings used in prior BCI studies (12). This change resulted in a $10\%$ WER increase, suggesting that relative positional information is important for focusing on local features relevant to phoneme decoding. We next removed time-masking, and restored regularization hyperparameters (dropout, input dropout, white noise, baseline shift) to the baseline levels used for the GRU (Table 5). Removing time-masking increased WER by $18\%$. Since the Transformer model without time-masking was trained for 250 epochs, this indicates that the performance benefits of the time-masked Transformer over the baseline GRU (which was trained for 73 epochs) cannot be attributed to extended training time alone. Finally, we replaced the Transformer architecture with the GRU architecture. When training the time-masked GRU, we used the same regularization hyperparameters as the time-masked Transformer (Table 6) and trained for 250 epochs. The time-masked GRU only performed $4\%$ worse, whereas the baseline GRU performed $20\%$ worse, than the time-masked Transformer suggesting that time-masking is a generally useful augmentation for neural speech decoding across network architectures.

## 4.5 Optimizing the time-masked GRU with non-overlapping inputs

Results from the ablation study suggest that the time-masked GRU performs competitively with the time-masked Transformer. However, a key benefit of the Transformer is that it requires lower memory resources and is faster to calibrate. We hypothesized that the computational efficiency benefits of the Transformer are related to its ability to effectively process non-overlapping inputs, reducing processing redundancy. While prior work showed the baseline GRU performs optimally with overlapping inputs (41), we tested if this was also the case for the time-masked GRU. To investigate this, we trained the time-masked GRU with non-overlapping inputs by setting both the window length and stride length to 80ms (Section 3.4). We observed that performance was significantly worse with non-overlapping inputs, with validation WER when using a 3-gram LM increasing from $17.85 \pm 0.13\%$ to $19.66 \pm 0.14\%$ ($p < 0.05$; two-sided independent t-test, $N = 4$ seeds). To account for the possibility that a smaller hidden state would be more optimal with the smaller, non-overlapping inputs, we lowered the hidden state size from 1024 to 768 or 512, but this further worsened performance. Thus, consistent with Willett et al. (41), our results suggest that the GRU performs optimally with overlapping inputs even with time-masking. We leave it to future work to further investigate whether a smaller GRU architecture can perform competitively with the Transformer.

Table 4: Ablation study results for the Time-Masked Causal Transformer. Values represent word error rates (WER) on the validation data with a 3-gram LM as a percentage (mean $\pm$ SEM across seeds). 10 seeds were run for the original model, and 4 seeds were run for the ablations.

| Ablation | Validation WER (%) |
|---|---|
| Time-masked Transformer | $17.15 \pm 0.15$ |
| No Log Transform | $17.85 \pm 0.15$ |
| No Learning Rate Scheduler | $17.85 \pm 0.21$ |
| No T5 Positional Encoding | $18.89 \pm 0.06$ |
| No Time-Masking | $20.17 \pm 0.2$ |
| Transformer $\rightarrow$ GRU | $17.85 \pm 0.12$ |

# 5 Discussion

There are three noteworthy limitations to our results. First, all results are on a single participant. At the time of writing this manuscript, there were no other open-source microelectrode array datasets for speech neuroprostheses, and evaluation on one participant is standard of the field. However, it is still important to evaluate the effectiveness of time-masked Transformers and DietCORP across multiple participants. Second, our use of beam search allows previously decoded text to be revised, which complicates integration with text-to-speech systems and may also provide a suboptimal experience for users if textual revisions are significant. Third, even the smallest language modeling setup used in this study, the 3-gram LM, requires about $\sim 60\,\mathrm{GB}$ of CPU memory to operate. Such a large memory requirement makes it challenging to run the current decoding algorithm on local, resource-constrained devices.

In summary, this work delivers three practical advances for the Brain-to-Text Benchmark '24 and, more broadly, for the development of speech neuroprostheses. First, we showed that large amounts of time-masking serves as a highly effective data augmentation method for improving neural speech decoding accuracy. Second, replacing the baseline GRU with a Transformer provides substantial reductions in computational costs, making the Transformer an ideal architecture for on-device test-time adaptation. Third, we introduced DietCORP, a simple, lightweight, and fast method that utilizes multiple time-masked augmentations of the same trial to effectively adapt the Transformer at test-time. Together, these innovations lower word error rate while cutting resource demands, moving us closer to building robust, on-device speech neuroprostheses that restore fluent communication for patients with severe paralysis.

There are several exciting avenues for further improving speech neuroprostheses. First, future research can investigate avenues to reduce memory costs associated with the n-gram LM or explore alternative strategies to integrate text-based knowledge. For instance, Feng et al. [18] and a recent study [1] project the outputs of a GRU or Transformer to the token space of an LLM for text decoding, obviating the need for the n-gram LM. Second, a recent trend in automatic speech recognition is to build a single model that can operate in both streaming and non-streaming modes [45]. Such an approach could be useful for speech neuroprostheses, since the user may have different preferences for accuracy versus latency depending on the context. Third, as more microelectrode array datasets are released, an emerging research question is whether integrating microelectrode array recordings across multiple participants can be used to boost performance. This question has been explored with other neural recording modalities [10]. Finally, future work should explore achieving similar accuracy levels without using beam search, so that previous outputs cannot be revised for easier integration with text-to-speech systems [25].

We hope that our study encourages future speech neuroprostheses benchmarks, such as the Brain-to-Text Benchmark '25, to offer a more holistic evaluation criteria for decoding algorithms beyond only accuracy. By evaluating decoding algorithms along multiple criteria such as ability to decode in real-time, computational costs, and integration with existing test-time adaptation methods, these benchmarks can more effectively spur innovation and assist paralyzed patients with communication impairments.

## Acknowledgements

This work was supported by the following awards. To JCK: NSF CAREER 1943467, NIH DP2NS122037, NIH R01NS121097. To EF: NIH T32NS115753.

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

## A  Computational resources

The majority of our results were generated using an Ubuntu server with three Nvidia GeForce RTX 3090 GPUs and an AMD Ryzen Threadripper 3960X 24-Core CPU with 125 GiB of memory. For the 5-gram LM and LLM fine-tuning results, we used the "g6e.16xlarge" Amazon EC2 instance with one Nvidia L40S GPU and an AMD EPYC 64-core CPU with 512 GiB of memory.

## B  Code availability

Code is available at the following link: `https://github.com/ebrahimfeghhi/transformers_with_dietcorp`.

## C  FLOPS

We computed the number of floating point operations per second (FLOPS) for the baseline GRU and Transformer. When computing FLOPS for the baseline GRU, we included the day-specific linear layer, the 5 GRU layers, and the output linear layer. When computing FLOPS for the Transformer, we included the patch embedding linear layer, the 5 Transformer layers, and the output linear layer. We removed all regularization for this calculation, and computed FLOPS using 10 seconds of input, and then divided by 10.

## D  Hyperparameters

We list the hyperparameters for the baseline GRU in Table 5 and for the time-masked Transformer in Table 6.

## E  Language model and beam search

We used the 3-gram and 5-gram language models (LMs) provided by Willett et al. [41], and use the same language model decoding setup as the Willett et al. [41]. We provide a brief description of the language model and beam search process here, with additional details provided in the original study.

The n-gram LM was converted to a weighted finite state transducer (WFST) [31]. This language model WFST, referred to as the grammar WFST, was composed with two additional WFSTs representing the mapping from phonemes to words (lexicon WFST) and the mapping between phonemes and the CTC blank token to model logits (token WFST). To compute the word error rate (WER), we applied beam search on the composed WFST graph to obtain the most likely text sequence from

Table 5: Baseline GRU hyperparameters. Window length indicates the number of stacked neural time bins fed as input per GRU step, and stride size indicates the rate at which the GRU outputs phonemes.

| Hyperparameter | Value |
| --- | --- |
| Window Length | 32 (640 ms) |
| Stride Size | 4 (80 ms) |
| Layers | 5 |
| Hidden Size | 1024 |
| LR | 0.02 |
| Epochs | 73 |
| White Noise | 0.8 |
| Baseline Shift | 0.2 |
| L2 Decay | $1e-5$ |
| Dropout | 0.4 |
| Input Dropout | 0.0 |
| Gaussian Smooth Kernel Size | 20 |
| Gaussian Smooth $\sigma$ | 2.0 |

Table 6: Transformer architecture and training hyperparameters. Feed-forward network (FFN) multiplier indicates the increase in dimensionality in the FFN module. The values $N$ and $M$ represent the number of time-masks and max mask length as a percentage of trial duration, respectively.

| Hyperparameter | Value |
|---|---|
| Patch Length | 5 (100 ms) |
| Patch Width | 256 |
| Transformer Dim | 384 |
| Layers | 5 |
| Heads | 6 |
| Dim Head | 64 |
| FFN Multiplier | 4 |
| LR | 0.001 |
| White Noise | 0.2 |
| Baseline Shift | 0.05 |
| $N$ | 20 |
| $M$ | 7.5 |
| Epochs | 250 |
| Dropout | 0.35 |
| Input Dropout | 0.2 |
| L2 Decay | $1e-5$ |
| Gaussian Smooth Kernel Size | 20 |
| Gaussian Smooth $\sigma$ | 2.0 |

the GRU/Transformer. We denote the GRU/Transformer as the "encoder" for this section. The probability of a given beam (i.e., one possible decoded transcription), $b$, was computed as a weighted sum between the probability assigned by the encoder and the probability assigned by the n-gram language model:

$$\text{score}(b) = \alpha \cdot \log(P_{\text{enc}}(b)) + \log(P_{\text{ngram}}(b))$$

The beam with the lowest log probability was then selected as the final decoded transcription. Blank labels emitted by the encoder were additionally penalized by dividing their probability by a constant. When using the 3-gram LM, we used a beam size of 18, $\alpha = 0.8$, and set the blank penalty to log(2). For the 5-gram LM, we modified the blank penalty to log(7). All decoding hyperparameters were the same as in the original Willett et al. [41] study.

When using the 5-gram LM, two additional decoding steps were performed. First, a second pass was applied with an unpruned 5-gram LM. The second pass replaces the original LM beam scores with updated, more accurate scores from the unpruned LM. Second, the top $K$ beams were returned, and an LLM (OPT 6.7B [47]) was used to score these top $K$ beams. The updated score for each beam was then:

$$\text{score}(b) = \alpha \cdot \log(P_{\text{enc}}(b)) + \beta \cdot \log(P_{\text{ngram}}(b)) + (1 - \beta) \log(P_{\text{opt}}(b))$$

Following Willett et al. [41] we set $K = 100$, $\beta = 0.5$, and ran OPT 6.7B in 8-bit precision mode.

## F  Linderman Lab GRU

The Linderman Lab bidirectional GRU modified the original baseline GRU architecture to include the following stack of layers after the final (5th) GRU layer.

```
self.fc_decoder_out = nn.Sequential(
    nn.LayerNorm(h * 2),
    nn.Dropout(p=dropout),
    nn.Linear(h * 2, h * 2),
    nn.SiLU(),
```

```
        nn.LayerNorm(h * 2),
        nn.Dropout(p=dropout),
        nn.Linear(h * 2, h * 2),
        nn.SiLU(),
        nn.Dropout(p=dropout),
        nn.Linear(h * 2, nclasses + 1),
    )
```

Where *h* refers to the number of hidden units in the GRU (1024), *dropout* refers to the dropout probability (0.4) and *nclasses* refers to the number of phonemes (40). Beyond changes to the model architecture, the GRU was trained using a linear learning rate scheduler, starting from 0.025 and ending at 0.0005 for 20,000 batches ($\sim 146$ epochs). An input dropout layer, with the dropout value set to 0.3 was also included before the day-specific linear layer. All other hyperparameters were the same as the baseline GRU model.

## G   Large language model fine-tuning

To perform LLM fine-tuning, we first generated decoded transcripts using 10 seeds of the time-masked Transformer and the 5-gram LM setup for the entire training and validation set. We then fine-tuned Llama 3.1 8B to predict the ground-truth transcript given the 10 decoded sentences for each trial. Fine-tuning was performed by "masking" the ground-truth transcript and training the LLM to predict the masked tokens with cross-entropy loss. The LLM was fine-tuned with QLoRA ([14]) via the Unsloth package ([13]). We used the default hyperparameters provided by Unsloth for Llama 3.1 8B, which are listed in the table below. A prompt, also provided below, was used to guide the LLM for each trial. As no hyperparameter tuning was performed, our procedure consisted of fine-tuning the LLM first on the validation set, then on the training set, and finally once more on the validation set. We report results from a single LLM fine-tuning seed.

Table 7: Hyperparameters for model training. Lora-specific parameters are listed at the bottom.

| Hyperparameters | Values |
| --- | --- |
| Warmup Steps | 5 |
| Epochs | 1 |
| Learning Rate | $2 \times 10^{-4}$ |
| Weight Decay | 0.01 |
| Learning Rate Scheduler | Linear |
| Batch Size | 16 |
| Lora Alpha | 16 |
| Lora Dropout | 0 |
| Lora Rank | 16 |

LLM Prompt: *Your task is to perform automatic speech recognition error correction. Below are multiple candidate transcriptions of the same utterance. These candidates were decoded from neural activity and may contain errors. Based on the candidates, produce the single most accurate, coherent, and grammatical transcription of the original utterance. Focus on key differences between candidates that change meaning or correctness, and avoid repetitive or nonsensical phrases. Respond with only the final corrected transcription—no explanations or extra text.*

## H   Beam search decoding time

We measured the average time required for beam search decoding using the 3-gram language model, excluding the time to generate logits. With the baseline GRU, the average decoding time across 1200 test trials was $0.056 \pm 0.042$ seconds per trial (mean $\pm$ standard deviation). In contrast, the time-masked Transformer was over 3 times faster, achieving an average decoding time of $0.017 \pm 0.011$ seconds. One potential explanation is that the Transformer outputs logits at 10 Hz, compared to 12.5 Hz for the GRU, resulting in fewer decoding steps. However, the time-masked GRU, which shares the same output resolution as the baseline GRU, achieved a decoding speed of $0.041 \pm 0.034$ seconds, meaning the reduced output temporal resolution does not fully account for the faster decoding times.

Table 8: Performance under different masking strategies across 4 seeds. $N$ is the number of masks, and $M$ is the max mask length as a percentage of trial duration.

| $N$ and $M$ | Validation WER (%) |
|---|---|
| 20 and 15 | $26.16 \pm 2.11$ |
| 20 and 12.5 | $20.27 \pm 0.33$ |
| 20 and 10.0 | $17.74 \pm 0.28$ |
| 20 and 8.5 | $17.15 \pm 0.27$ |
| 20 and 7.5 | $17.08 \pm 0.26$ |
| 20 and 5.0 | $17.96 \pm 0.31$ |
| 20 and 2.5 | $20.81 \pm 0.29$ |
| 20 and 1.5 | $23.14 \pm 0.11$ |
| 10 and 15.0 | $17.08 \pm 0.18$ |
| 10 and 10.0 | $17.31 \pm 0.31$ |
| 30 and 7.5 | $18.14 \pm 0.33$ |
| 30 and 5.0 | $17.5 \pm 0.23$ |

## I  Performance with other masking strategies

We report validation WER across different masking ratios for the Transformer when using the 3-gram LM (Table 8). We set $N = 20$ and $M = 7.5$ throughout the study.

We additionally experimented with masking entire channels (channel-masking) instead of time-masking. In order to do so, we applied $N$ channel-masks for each microelectrode array. For each mask, we selected a starting channel $i$. Then, we masked (replaced with 0) the closest $p$ channels from channel $i$, where "closest" was quantified by the euclidean distance from channel $i$. $p$ was uniformly sampled between 0 and $M \times 64$ (there were 64 channels per microelectrode array), and $M$ could range from 0 to 1. Our preliminary results suggested that channel masking was not as effective as time-masking on this dataset, either when applied alone or in combination with time-masking.

## J  Calculating average masking percentage

Refer to Algorithm 1 for the full variable definitions. Here we derive the expected fraction of patches that are masked. Given that each mask length is uniformly sampled between 0 and $F$, the expected mask length is $F/2$. Ignoring boundary effects, the probability that a given patch $I$ is not masked by a given mask $N_j$ is:

$$P\big(\text{unmasked}_{I,N_j}\big) \approx 1 - \frac{\frac{F}{2}}{L} = 1 - \frac{F}{2L} = 1 - \frac{\lfloor ML \rfloor}{2L} \approx 1 - \frac{M}{2}$$

Then, the probability that a patch $I$ is unmasked after all masks are applied is:

$$P\big(i \text{ unmasked}\big) = \prod_{k=1}^{N} P\big(\text{unmasked}_{i,N_k}\big) = \prod_{k=1}^{N}\left(1 - \tfrac{M}{2}\right) = \left(1 - \tfrac{M}{2}\right)^{N}.$$

To find the proportion of masked patches, we take the complement. Using the linearity of expectation, we find that the expected proportion of masked patches is:

$$\mathbb{E}\big[\text{fraction of masked patches}\big] = \frac{1}{L}\sum_{i=1}^{L}\Big[1 - \left(1 - \tfrac{M}{2}\right)^{N}\Big] = 1 - \left(1 - \tfrac{M}{2}\right)^{N}.$$

For our given parameters, we approximate that $53.44\%$ of patches are masked on average for each trial.

