# OpenReview forum: "Time-Masked Transformers with Lightweight Test-Time Adaptation for Neural Speech Decoding"
_NeurIPS.cc/2025/Conference — NeurIPS 2025 poster_

### Official Review · Reviewer_YQY9 · 2025-06-22

**Clarity:** 3
**Significance:** 3
**Originality:** 3
**Rating:** 5
**Confidence:** 4

**Summary:**

This paper proposes three improvements over the baseline algorithm in the brain-to-speech benchmark:

1. Time masking to prevent overfitting
2. Replace GRU with a transformer to reduce the memory footprint
3. Test-time adaptation (TTA) via entropy minimization

With these improvements, the word error rate (WER) reduces from 15.39 to 12.27, which is quite significant compared to previous attempts. The results of TTA also look promising as a novel strategy to mitigate the non-stationarity problem inherent to many BCI decoders.

**Questions:**

1. Minimize in-session training time is a key objective for a clinically viable BCI. The transformer takes 600 epochs to train, which is ~2.3hr, while the original GRU baseline requires ~0.5hr.  Can you show how the transformer+maksing, gru+masking, and the gru baseline’s WER vary with training time? This can help the researchers pick the best model for the amount of time they have in session for training models.
2. The baseline GRU’s parameters were optimized for the overlapping inputs and may not be optimal for the time-masked non-overlapping inputs. To really support the claim that the transformer uses less memory than the GRU, both models’ hyperparameters need to be optimized with the time-masked non-overlapping inputs.
3. Can you run decoding with the 5gram LM with cloud computing resources?
4. In Sec 4.3, if performance is measured in WER, is the difference between CORT and No CORT still similar?

**Ethical Concerns:**

["NO or VERY MINOR ethics concerns only"]

**Final Justification:**

Thanks for the update. With the 8.01 WER when using 5-gram, and the stable performance of DietCORP, I believe this paper makes a very valuable contribution to brain-to-text BCI decoding algorithms.

One minor point is that calling the 5-gram+LLM rescoring offline isn't accurate. This decoding approach was used here for online decoding. The rescoring step is quick enough that it doesn't cause a significant delay. I suggest naming the metric in Tab 1. after the language models you use.

**Limitations:**

yes

**Paper Formatting Concerns:**

There are no line numbers; it might be compiled using the "preprint" style.

**Quality:**

3

**Strengths And Weaknesses:**

Strength:

- Many people have tried to replace the GRU baseline in the brain-to-text benchmark without success. This paper identifies the core problem (overfitting) and finds a simple solution (time masking).
- The first to apply TTA with entropy minimization to the BCI problems, promising results.
- The paper is clearly written and easy to follow. Experiments are well designed to support the claims.

Weakness:

- The major benefits of using a transformer are less GPU memory usage and faster recalibration. But they are very marginal: 5GB to 3GB, and 17ms to 13ms. Even though the per-epoch training is faster, the overall training actually takes longer due to more training epochs.
- The GRU’s hyperparameters are not re-optimized for the time-masked non-overlapping inputs. So it’s not fully convincing that the transformer is more efficient than the GRU.
- Though testing with a 5gram LM requires significantly more computing resources, all other brain-to-text benchmark methods report their results on the 5gram LM. The authors need to show their results with a 5gram LM to really convince the field that a masked transformer is a better approach.

---

> ### Author Rebuttal · Authors · 2025-07-27
>
> Thank you for the feedback. We appreciate that the Reviewer finds our paper easy to follow and finds the use of time masking to reduce overfitting interesting. We provide a one sentence response for each question and weakness, with additional details provided below. Tables are provided at the end of the text.
>
> **General Note**: We realized after submission that all the entries in Willett et al. 2025 [b] used a bidirectional GRU. A bidirectional GRU can only decode speech once the entire trial has finished, and roughly doubles memory usage, training times, and calibration times relative to the unidirectional GRU. All our models are either unidirectional GRUs or causal Transformers which only rely on present and past neural activity to generate outputs.
>
> **Training time takes longer when using time masking**: Our time-masked Transformer showed no drop in performance when trained for the same time as the entries which used GRUs.
>
> - *Additional Details:* We find that a $5$-layer time-masked Transformer trained for $250$ epochs (~1 hour of training time) exhibits slightly better validation WER than our original $7$-layer time-masked Transformer trained for either $600$ epochs or $250$ epochs. This may be because the smaller Transformer trained for less epochs has a lower validation CTC loss than the larger variant. We had previously only focused on validation phoneme error rate (PER), as done in Willett et al., 2023 [a]. This Transformer requires roughly the same amount of training time as the baseline bidirectional GRU and about half the training time as the Linderman Lab GRU, which was the best performing entry with only modifications to the GRU in Willett et al., 2025 [b] (Table 1).
>
> **Evaluate whether time-masked GRU can process non-overlapping inputs, and if this yields efficiency gains**: A time-masked GRU trained with non-overlapping inputs and a smaller hidden state size reduces parameter count by $35$ M relative to the original GRU architecture while maintaining competitive performance with our original time-masked GRU.
>
> - *Additional details:* Our original hypothesis was that the GRU would not perform optimally with non-overlapping inputs due to its lossy hidden state. Similar to Willett et al., 2023 [a], we found this was the case with the original GRU size. However, we found that performance with non-overlapping inputs improved when reducing the hidden size, suggesting that models with non-overlapping inputs generalize better when smaller. We will continue optimizing the GRU with non-overlapping inputs for efficiency gains, and update our manuscript to discuss how processing non-overlapping inputs may be a good heuristic to follow when designing lightweight architectures.
>
> **Results are not displayed with the 5-gram LM setup**: We have included results using the updated models with the 5-gram LM setup, which show that our models outperform strong baselines in a real-time decoding setting, and even match the performance of strong baselines which rely on bidirectional GRUs (Table 1).
>
> - *Additional details:* We have added results with the offline language modeling setup used by the entries in the competition (Willett et al., 2025 [b]). Specifically, the offline language modeling setup involved using a 5-gram LM, with a second pass using the 5-gram LM, and rescoring with OPT 6.7B. We refer to the setup where only the 3-gram LM is used with no second pass as the online LM setup. Results are displayed in Table 1.
>
> **Is the difference between CORT and No CORT similar with WER?**:  We have revised our test-time adaptation method to a computationally lightweight pseudo-label based approach, which was inspired by Fan et al., 2023 [c], and there is a substantial reduction in WER when applying our improved TTA method relative to no TTA (Table 2 and Table 3).
>
> - *Additional details:* While we did find a small reduction in WER when applying CORT relative to No CORT across days (average WER improvement across days shown in Figure 4, left), this improvement was small. We thus sought to examine if applying time masking and Transformers could provide efficiency gains to the pseudo-label based TTA method proposed by Fan et al., 2023 [c]. The method used by Fan et al., 2023, CORP, adapts a model at test-time by treating the language model (LM) refined model outputs as pseudo-labels, and then training the model with CTC loss using these pseudo-labels. Since the LM refined outputs are already being generated for real-world use, there is no additional cost to generate these pseudo-labels. However, CORP requires up to $200$ adaptation steps per test trial, setting a loss threshold hyperparameter, and including previous data to stabilize training. Here, we present "DietCORP", a lightweight version of CORP which only uses one adaptation step per trial and no previous data. Similar to CORT, DietCORP operates by generating several time masked augmentations of the same trial, and then it adapts the model to produce the same pseudo-label across all augmentations. We present results when applying DietCORP to a time-masked Transformer trained on $15$ days of data, and evaluated on the next $5$ competition days (Table 2), and a time-masked Transformer trained on $12$ days of data and evaluated on the next $8$ competition days (Table 3). Only the patch embedding module was adapted. DietCORP leads to very stable WER across days relative to when not applying it with minimal hyperparameter tuning.
>
> **Reductions in memory usage and calibration times are marginal**: With our updated TTA method, DietCORP, we reduce peak GPU memory usage by $81$% and calibration time by $82$% relative to the bidirectional GRU.
>
> - *Additional details:* DietCORP only requires $1.04$ GiB of peak allocated GPU memory usage and $29.3$ ms to calibrate per trial using the Transformer. When applying our TTA method to the unidirectional GRU, it requires $2.8$ GiB of peak allocated GPU memory usage and $76.9$ ms to calibrate per trial. These values further increase to $5.6$ GiB and $163.0$ ms when using the bidirectional GRU. We will also include comparisons when using CORP in the updated manuscript, where DietCORP applied to the Transformer exhibits even larger computational gains. We accept that the reductions in memory usage and training times are perhaps marginal for offline training, where models are often trained using large servers while the neuroprosthesis is not in use. However, we believe these efficiency gains are important in the context of test-time adaptation, where calibration is performed on-device in the background while the patient is using the neuroprostheses.
>
>
> ***
> **Tables**
>
> Table 1: We show results with the $5$-layer time-masked Transformer trained for $250$ epochs (Transformer - Ours) as well as comparisons with the baseline provided by Willett et al., 2023 [a] and the Linderman Lab GRU discussed in Willett et al., 2025 [b]. We run the baseline and Linderman Lab GRU in unidirectional mode, and show metrics in bidirectional mode in parenthesis when available.
>
>
> | Metric                    | Transformer - Ours | Baseline GRU       | Linderman Lab GRU   |
> |---------------------------|-------------------:|-------------------:|--------------------:|
> | Online LM Setup WER %    | 12.03 ± 0.24        | 15.25 ± 0.16        | 15.03 ± 0.19         |
> | Offline LM Setup WER %  | 8.01 ± 0.23         | 11.12 ± 0.13 (9.76) | 10.77 ± 0.13 (8.0)   |
> | Parameters (M).        | 9.4                 | 56.7 (135.4)        | 58.8 (143.8)         |
> | Training time (hours)   | ~1                  | ~0.5  ( ~1)         | ~1  ( ~2)            |
>
>
> Table 2: DietCORP is our computationally lightweight version of CORP, a test-time adaptation method proposed by Fan et al., 2023 [c]. Values are word error rate (mean ± SEM across $N=4$ seeds) computed on $5$ held-out competition days using a model trained on the previous $15$ days.
>
> | Days since last training day | No DietCORP | DietCORP  |
> |--------:|----------------:|----------------:|
> | 7       |  22.74 ± 0.26        | 21.14 ± 0.32        |
> | 13     |  25.48 ± 0.28        | 21.23 ± 0.21       |
> | 19     |  25.43 ± 0.33       | 21.40 ±  0.22       |
> | 28     |  30.13 ± 0.33     | 21.93 ±  0.65       |
> | 30     |  32.59 ± 0.52     | 22.26 ±  0.60        |
>
>
> Table 3: Values are word error rate (mean ± SEM across $N=4$ seeds) computed on $8$ held-out competition days using a model trained on the previous $12$ days.
>
> | Days since last training day | No DietCORP | DietCORP  |
> |--------:|----------------:|----------------:|
> | 5       | 28.87 ± 0.47        | 26.97 ± 0.28          |
> | 12     | 34.50 ± 0.44       | 26.38 ± 0.30         |
> | 21     | 44.79 ± 0.34       | 27.57 ±  0.13        |
> | 28       | 53.29 ± 0.25        | 30.46 ± 0.41         |
> | 34       | 59.98 ± 0.31        | 29.70 ± 0.20         |
> | 40       | 52.71 ±  0.52        | 29.08 ± 0.38         |
> | 49       | 62.96 ± 0.40         | 31.35 ± 0.76         |
> | 51       | 66.47 ± 0.58         | 31.91 ± 1.25        |
>
>
> [a] Willett et al., 2023, A high-performance speech neuroprosthesis
>
> [b] Willett et. al, 2025, Brain-to-Text Benchmark '24: Lessons Learned
>
> [c] Fan et al., 2023, Plug-and-Play Stability for Intracortical Brain-Computer Interfaces: A One-Year Demonstration of Seamless Brain-to-Text Communication

---

### Official Review · Reviewer_VfR8 · 2025-07-01

**Clarity:** 2
**Significance:** 2
**Originality:** 2
**Rating:** 2
**Confidence:** 4

**Summary:**

This paper examines the application of masked Transformers in conjunction with test-time adaptation techniques for speech neuroprostheses. It addresses the real-time processing and memory-efficiency limitations of existing GRU-based models, as well as distribution shifts in neural signals over time. The core idea is to apply time masking to sequential neural activity during training to mitigate overfitting and to deploy a lightweight Transformer architecture that maximizes parallelism and reduces the memory footprint. At test time, the model adapts to ensure consistent predictions on time-masked inputs, thereby enhancing its robustness to changes in temporal distribution.

**Questions:**

- Would similar results be obtained using Dorsal 6v instead of Ventral 6v?
- Why is a Transformer architecture more effective than GRUs when paired with time masking?
- Are the default time masking parameters (N = 20, M = 0.075) optimal?

**Ethical Concerns:**

["NO or VERY MINOR ethics concerns only"]

**Final Justification:**

As stated in the introduction, the authors claim to propose a holistic approach, centered on the model architecture (including masking strategies). They explicitly note that GRUs tend to overfit in the early stages of training and that they process highly redundant inputs at each time step, which is a limitation.

However, when GRUs are applied bidirectionally, the WER differs by only 0.09%, as also reported in Author Final Remarks. This result suggests that the importance lies more in incorporating bidirectional context rather than in the GRU architecture or the masking strategy itself. Consequently, I believe the validity of the authors’ claim is weakened, and the proposed technique lacks sufficient supporting evidence. Furthermore, in the context of Reviewer FhGX’s comments, I find the claim of being a “holistic” technique to be over-hyped.

In conclusion, while the research topic and the additional experiments presented during the review process are impressive, due to the significant issues outlined above, I have decided to assign a final rating of reject.

**Limitations:**

See above

**Quality:**

2

**Strengths And Weaknesses:**

1. Claims And Evidence
    - Strengths
        - Figure 2 demonstrates that applying time masking significantly narrows the gap between training and validation CTC loss, indicating its effect in mitigating overfitting.
        - Table 7 shows the effectiveness of time masking across various choices of N and M.
        - Figure 3 presents a clear comparison between the Transformer and a baseline GRU, revealing simultaneous gains in both efficiency and accuracy under diverse conditions.
        - By relying solely on the Transformer model, without LLMs or ensemble strategies, the approach achieves optimized streaming performance.
    - Weaknesses
        - Time masking is effectively a form of data augmentation; however, the paper lacks a comparative evaluation against other augmentation methods, such as SpecAugment, additive white noise, and speed perturbation [a, b].
        - The GRU-based TeamCyber model [c] attains a WER of 8.26%, whereas the proposed method yields 12.27 ± 0.15%, a substantially worse result that calls into question its significance.
2. Methods And Evaluation Criteria
    - Weaknesses
        - Since time masking is a widely used technique, it is difficult to regard it as the paper’s main contribution. To make such a claim credible, the authors would need to benchmark it against alternative approaches (e.g., speckled masking [g]).
        - Beyond CORT, the paper does not compare with other established test-time adaptation methods (e.g., pseudo-label–based adaptation, sample-efficient entropy minimization, test-time normalization) [d, e, f].
        - The Method section lacks a detailed mathematical formulation for how test-time adaptation is applied; a more explicit derivation would strengthen reproducibility.
3. Theoretical Claims
    - Weaknesses
        - The paper provides limited theoretical justification for why Transformers are inherently more efficient and robust than GRUs and lacks experiments comparing alternative architectures or variants to demonstrate generality.
4. Experimental Designs Or Analyses
    - Weaknesses
        - An exploratory data analysis illustrating day-to-day distribution shifts in the neural signal data would have been informative.
        - The proposed method updates only the weights of the patch embedding module during online TTA while freezing all other layers. In contrast, the method in [18] performs partial fine-tuning across every layer; however, the authors did not evaluate a similar multi-layer adaptation strategy—an experiment that would be necessary to assess the impact of broader fine-tuning.
        - DConD-LIFT [c] achieved a WER of 5.77% by combining an RNN ensemble with an LLM; the proposed method’s performance appears to be far inferior. Although the paper cites memory usage as the key advantage of using a Transformer, it is unclear whether this benefit justifies the performance gap, and no supporting experiments are provided.
5. Others
    - Typographical error: “partiicpant” (Section 5: Limitations)

>
> [a] Park, Daniel S., et al. "Specaugment: A simple data augmentation method for automatic speech recognition." *arXiv:1904.08779* (2019).
>
> [b] Huh, Mina, Ruchira Ray, and Corey Karnei. "A Comparison of Speech Data Augmentation Methods Using S3PRL Toolkit." arXiv:2303.00510 (2023).
>
> [c] Willett, Francis R., et al. "Brain-to-Text Benchmark'24: Lessons Learned." *arXiv:2412.17227* (2024)
>
> [d] Wang, Qin, et al. "Continual test-time domain adaptation." *Proceedings of the IEEE/CVF Conference on Computer Vision and Pattern Recognition*. 2022.
>
> [e] Niu, Shuaicheng, et al. "Efficient test-time model adaptation without forgetting." *International conference on machine learning*. PMLR, 2022.
>
> [f] Lim, Hyesu, et al. "Ttn: A domain-shift aware batch normalization in test-time adaptation." ICLR (2023).
>
> [g] Keshtkaran, Mohammad Reza, and Chethan Pandarinath. "Enabling hyperparameter optimization in sequential autoencoders for spiking neural data." *Advances in neural information processing systems* 32 (2019).
>

---

> ### Author Rebuttal · Authors · 2025-07-29
>
> Thank you for the feedback. We appreciate that the Reviewer noted our focus on optimizing performance in a real-time streaming setting. We provide a one sentence response for each question and weakness, with additional details provided below. Tables are provided at the end of the text.
>
> **General Note**: We realized after submission that all the entries in Willett et al. 2025 [b] used a bidirectional GRU. A bidirectional GRU can only decode speech once the entire trial has finished, and roughly doubles memory usage, training times, and calibration times relative to the unidirectional GRU. All our models are either unidirectional GRUs or causal Transformers which only rely on present and past neural activity to generate outputs.
>
> **Low performance on benchmark**: When using the 5-gram language modeling setup used by the entries in the Brain-to-Text Benchmark ‘24, the time-masked Transformer achieves a mean WER of 8.01%, which ties the 2nd place entry (Linderman Lab GRU) despite using ~$15$ times less parameters and decoding in real-time. (Table 1).
>
> - *Additional Details:* We have added results (Table 1) with the offline language modeling (LM) setup used by the entries in Willett et al., 2025 [b]. Specifically, the offline language modeling setup involved using a 5-gram LM, with a second pass using the 5-gram LM, and rescoring with OPT 6.7B. We refer to the setup where only the 3-gram LM is used with no second pass as the online LM setup.
>
> **DConD-LIFT achieves 5.77% WER**: The primary performance gain in the first place entry, DConD-LIFT, stems from applying GPT 3.5, while their main improvement to the GRU itself (decoding diphones) achieves a WER of 8.06%.
>
> - *Additional Details:* DConD-LIFT (Li et al., 2023) shows remarkable accuracy gains by applying GPT 3.5 to an ensemble of bidirectional GRUs. However, since their submission requires access to cloud resources and is not streaming compatible, we believe a comparison with our method strictly based on accuracy is not entirely fair.
>
> **Benchmark time masking against other regularization strategies**: We have included a comparison with the Linderman Lab GRU which incorporates speckled masking (i.e. input dropout) in Table 1, and our original study also included comparisons with other regularization strategies (channel masking, additive white noise, baseline shifts).
>
> - *Additional Details*: We ran the Linderman Lab GRU, which uses specked masking (i.e. input dropout), in unidirectional mode. The time-masked Transformer performs substantially better than the Linderman Lab GRU in a real-time decoding setting (Table 1), and matches the performance of the bidirectional Linderman Lab GRU (Table 1). Furthermore, the baseline GRU from Willett et al., 2023 [a] used high amounts of additive white noise and baseline shift augmentations, providing a strong comparison with our proposed time masking augmentation. We also incorporated speckled masking (denoted as input dropout in our manuscript), additive white noise, and baseline shifts when training models with time masking. We mentioned these details in the Methods and Appendix, and will state them more clearly in the updated manuscript. Regarding SpecAugment, we note that SpecAugment consists of three parts: time masking, channel masking, and time warping on an audio spectrogram. Time masking was a major focus of our study, and we also provided preliminary using channel masking in Appendix Section 7.3. We did not implement time warping since it had a minor effect in the original SpecAugment study (Park et al., 2023 [e]), and we did not implement speed perturbation since we are working with data from one patient who speaks at a relatively constant rate.
>
> **Comparison with existing test-time adaptation (TTA) methods**: The main TTA method in the field of communication neuroprostheses is continual online calibration with pseudo-labels (CORP) (Fan et al., 2023 [c]), and in our updated manuscript we present a computationally lightweight version of CORP.
>
> - *Additional Details:* In CORP, the model is adapted at test-time by treating the language model refined model outputs as pseudo-labels. CORP requires up to $200$ adaptation steps per test trial, setting a loss threshold hyperparameter, and including previous data to stabilize training. We propose "DietCORP", a lightweight version of CORP which only requires one adaptation step per test trial and does not require maintaining previous data. Similar to CORT, DietCORP operates by generating several time masked augmentations of the same trial, and then it adapts the model to produce the same pseudo-label across all augmentations. We present results when applying DietCORP to a time-masked Transformer trained on $15$ days of data, and evaluated on the next $5$ competition days (Table $2$), and a time-masked Transformer trained on $12$ days of data and evaluated on the next $8$ competition days (Table 3). DietCORP leads to stable WER across days relative to when not applying it. DietCORP only requires $1.04$ GiB of peak allocated GPU memory usage and $29.3$ ms to calibrate per trial using the Transformer, versus $5.6$ GiB and $163$ ms to calibrate when using the bidirectional GRU. We will provide direct comparisons with CORP in the updated manuscript.
>
> **Only the patch embedding module is updated for TTA**: We only updated the patch embedding module because it requires less resources, which is important for edge devices, and we will show additional results when adapting a larger fraction of the weights in the updated manuscript.
>
> **Mathematical Formulation for TTA**: We will provide a detailed mathematical formulation of our updated TTA method in the manuscript.
>
> **Theoretical investigation into why Transformers are more efficient than GRUs**: We hypothesize that Transformers are more efficient because they were optimized to process non-overlapping inputs, and show efficiency gains with a GRU that processes non-overlapping inputs.
>
> - *Additional Details:* When optimizing the time-masked GRU to process non-overlapping inputs, we obtain competitive performance with a GRU that uses less than half the parameters as the baseline GRU ($21$ M versus $56$ M). We will provide a discussion of this in the updated manuscript.
>
> **Exploratory analysis of day-to-day shifts in neural activity**: We will discuss previous work (Willett et al., 2021 [f]) which shows that day-to-day shifts in neural activity can be represented as rotations from the original space into a new space in our updated manuscript.
>
> **Optimal hyperparameters for time masking**: We show results with other time masking hyperparameters in Appendix Section 7.3, and will reference this more clearly in the updated manuscript.
>
> **Dorsal 6v**: The time-masked Transformer performs better than the baseline GRU when using only Dorsal 6v, which we will include in the updated manuscript.
>
> ***
> **Tables**
>
> Table 1: We show results with the time-masked Transformer trained for 250 epochs (Transformer - Ours) as well as comparisons with the baseline provided by [a] and the Linderman Lab GRU discussed in [b]. We run the baseline and Linderman Lab GRU in unidirectional mode, and show metrics in bidirectional mode in parenthesis when available.
>
>
> | Metric                    | Transformer - Ours | Baseline GRU       | Linderman Lab GRU   |
> |---------------------------|-------------------:|-------------------:|--------------------:|
> | Online LM Setup WER %   | 12.03 ± 0.24        | 15.25 ± 0.16        | 15.03 ± 0.19         |
> | Offline LM Setup WER %  | 8.01 ± 0.23         | 11.12 ± 0.13 (9.76) | 10.77 ± 0.13 (8.0)   |
> | Parameters (M)          | 9.4                 | 56.7 (135.4)        | 58.8 (143.8)         |
> | Training time (hours) | ~1                  | ~0.5  ( ~1)         | ~1  ( ~2)            |
>
>
> Table 2: Values are word error rate (mean ± SEM across N=$4$ seeds) computed on $5$ held-out competition days using a model trained on the previous $15$ days.
>
> | Days since last training day | No DietCORP | DietCORP  |
> |--------:|----------------:|----------------:|
> | 7       |  22.74 ± 0.26        | 21.14 ± 0.32        |
> | 13     |  25.48 ± 0.28        | 21.23 ± 0.21       |
> | 19     |  25.43 ± 0.33       | 21.40 ±  0.22       |
> | 28     |  30.13 ± 0.33     | 21.93 ±  0.65       |
> | 30     |  32.59 ± 0.52     | 22.26 ±  0.60        |
>
> Table 3: Values are word error rate (mean ± SEM across N=$4$ seeds) computed on $8$ held-out competition days using a model trained on the previous $12$ days.
>
> | Days since last training day | No DietCORP | DietCORP  |
> |--------:|----------------:|----------------:|
> | 5       | 28.87 ± 0.47        | 26.97 ± 0.28          |
> | 12     | 34.50 ± 0.44       | 26.38 ± 0.30         |
> | 21     | 44.79 ± 0.34       | 27.57 ±  0.13        |
> | 28       | 53.29 ± 0.25        | 30.46 ± 0.41         |
> | 34       | 59.98 ± 0.31        | 29.70 ± 0.20         |
> | 40       | 52.71 ±  0.52        | 29.08 ± 0.38         |
> | 49       | 62.96 ± 0.40         | 31.35 ± 0.76         |
> | 51       | 66.47 ± 0.58         | 31.91 ± 1.25        |
>
> [a] Willett et al., 2024, A high-performance speech neuroprosthesis
>
> [b] Willett et. al, 2025, Brain-to-Text Benchmark '24: Lessons Learned
>
> [c] Fan et al., 2023, Plug-and-Play Stability for Intracortical Brain-Computer Interfaces: A One-Year Demonstration of Seamless Brain-to-Text Communication
>
> [d] Li et al., 2024, Brain-to-Text Decoding with Context-Aware Neural Representations and Large Language Models
>
> [e] Park et al., 2019, SpecAugment: A Simple Data Augmentation Method for Automatic Speech Recognition
>
> [f] Willett et al., 2021, High-performance brain-to-text communication via handwriting

---

> > ### Comment · Reviewer_VfR8 · 2025-08-05
> >
> > We thank the authors for their detailed responses and additional experiments. Many of my initial concerns have been sufficiently addressed. In particular, it is now clear that the proposed method offers substantial advantages in online settings and computational efficiency, which strengthens the motivation for this work. However, several fundamental issues remain.
> >
> > In terms of clarity and significance, while the method achieves performance comparable to the Linderman Lab GRU and offline language model setups, the paper does not clearly articulate the benefits in the online LM setting. The focus remains mainly on the model architecture rather than the practical implications of the online formulation. This limits the clarity and may lead to an overstatement of the method’s contributions. Furthermore, the claim that Transformers outperform GRUs is not convincingly supported by the presented evidence.
> >
> > Regarding quality, questions remain about the use of Dorsal 6v and the role of bidirectional GRUs. Additionally, while time masking is central to the proposed approach, the paper lacks a hyperparameter sensitivity analysis. Since such methods are known to be sensitive to masking parameters, this omission weakens the empirical grounding of the results.
> >
> > That said, I acknowledge that the authors have enhanced the originality of their work through the rebuttal, particularly by clarifying its relevance to streaming neuroprosthetic systems. Therefore, I am raising my score by one level.

---

> ### Author Response · Authors · 2025-08-05
>
> We thank the reviewer for the response and are very appreciative of the increase in score. We would like to make several clarifications regarding the points the reviewer brought up, which we detail below. Reviewer comments are in quotes.
>
> **Note**: In our original response, we used the term *offline LM setup* when using the 5-gram LM, second pass with the 5-gram LM, and rescoring with OPT 6.7B, and the term *online LM setup* to refer to only using the 3-gram LM. We used these terms because they were used by Willett et al., 2023. However, as pointed out by Reviewer YQY9, later work has used the “offline” setup for online experiments. We thus replace the term *offline LM setup* with *5-gram + 2nd pass + OPT setup* and the term *online language modeling setup* with *3-gram setup* in future responses.
>
> "**In terms of clarity and significance, while the method achieves performance comparable to the Linderman Lab GRU and offline language model setups, the paper does not clearly articulate the benefits in the online LM setting.**"
>
> We would like to clarify that the causal, real-time decoding time-masked Transformer achieves performance comparable to the non-causal, bidirectional Linderman Lab GRU. When comparing both models in a real-time decoding setting, our model performs roughly $25$% better than the Linderman Lab GRU when using the *5-gram + 2nd pass + OPT setup* (originally referred to as offline LM setup, see note above) and the *3-gram setup* (originally referred to as online LM setup). These benefits were stated for both of these setups in Table 1 in our previous response.
>
>
> "**The focus remains mainly on the model architecture rather than the practical implications of the online formulation. This limits the clarity and may lead to an overstatement of the method’s contributions.**"
>
> The improvement on model architecture, specifically replacing the Transformer with the GRU, is only out of three of our core contributions. We also show that time masking is a highly effective augmentation on this benchmark and propose a computationally lightweight while effective test-time adaptation method by leveraging our two previous contributions. These contributions together offer a neural speech decoding algorithm that improves accuracy while reducing computational costs associated with test-time adaptation. We thus believe our contributions offer clear practical implications that are well-supported by the empirical data presented.
>
> "**Furthermore, the claim that Transformers outperform GRUs is not convincingly supported by the presented evidence.**"
>
> In the manuscript we submitted to NeurIPS we acknowledged that switching from Transformers to GRUs only leads to a small accuracy gain. Therefore, our focus is not on the fact that Transformers outperform GRUs in terms of accuracy. The primary benefit of Transformers over GRUs is that they require less memory to train and have faster calibration speeds, which makes them ideal for test-time adaptation. As noted in our previous response, applying DietCORP to Transformers leads to an $81$% reduction in peak GPU memory usage and an $82$% reduction in calibration time relative to applying DietCORP to bidirectional GRU.  When comparing to a unidirectional GRU, the Transformer still reduces peak memory usage by $63$% and calibration times by $62$%. We therefore believe that our claims about the benefit of the Transformer, specifically its suitability for test-time adaptation, are well supported by the present evidence.
>
> "**Regarding quality, questions remain about the use of Dorsal 6v and the role of bidirectional GRUs.**".
>
> We ran analyses with Dorsal 6v only and found that time-masked Transformer maintains its performance advantage over the baseline GRU, as noted in our previous response. We are happy to perform additional analyses regarding Dorsal 6v if the reviewer has additional questions. As we stated in our *General Note*, bidirectional GRUs were used by the entries in Willett et al., 2025, and we make this explicit because our proposed decoding algorithm operates in real-time whereas bidirectional GRUs must wait for the entire trial to finish. Therefore, the role of bidirectional GRUs is mainly to compare our performance in the more challenging, real-time (streaming) setting against previous entries that operated in the non-real-time (non-streaming) setting. We agree that in the ideal case there would be a separate benchmark track for streaming decoding algorithms and non-streaming algorithms.
>
> "**Additionally, while time masking is central to the proposed approach, the paper lacks a hyperparameter sensitivity analysis. Since such methods are known to be sensitive to masking parameters, this omission weakens the empirical grounding of the results.**"
>
> As we stated in our previous response under the title **Optimal hyperparameters for time masking**, we included this information in our original submission in Appendix Section $7.3$.

---

> ### Comment · Reviewer_VfR8 · 2025-08-05
>
> I thanks the authors for their additional response. As the authors pointed out, the paper does include experiments related to hyperparameters. However, the explored range appears to be distributed on both sides of the spectrum, making it difficult to tune effectively. Moreover, the range is too limited to observe clear trends, and it remains challenging to identify optimal values. This remains a limitation of the current work.

---

> ### Author Response · Authors · 2025-08-05
> **Time Masked Hyperparameters**
>
> We will show an increased hyperparameter tuning range in the updated manuscript. We address each point below.
>
> "**However, the explored range appears to be distributed on both sides of the spectrum, making it difficult to tune effectively.**"
>
> We are unsure what both sides of the spectrum means. If it means that we tuned the hyperparameters both up and down, we are unsure why this is an issue. We are happy to follow a procedure suggested by the reviewer to tune the hyperparameters.
>
> "**Moreover, the range is too limited to observe clear trends, and it remains challenging to identify optimal values.**"
>
> In our original submission we showed performance across 6 time masking hyperparameters. We found validation PER was stable for a range of time masking choices except when lowering the masking percentage to $2.5\%$ or when increasing the number of masks to $30$. We believe the relative robustness of our method to a variety of time masking hyperparameter is an advantage, rather than the lack of a clear trend being a disadvantage.
>
> "**This [lack of hyperparameter tuning for time masking] remains a limitation of the current work.**"
>
> Increasing our time masking hyperparameter tuning range would only likely increase the accuracy of the time-masked Transformer. Therefore, while we definitely agree with the reviewer that more extensive hyperparameter tuning is beneficial, we believe it does not limit our current claims.

---

> > ### Comment · Reviewer_VfR8 · 2025-08-07
> >
> > Thank you for presenting your perspective. Regarding the hyperparameters, my concern is the inherent difficulty of tuning, as both values are continuous and must be adjusted jointly. I have no further questions. Additionally, taking into account the perspective of another reviewer, I will revise my rating accordingly in the final justification.

---

> > > ### Author Response · Authors · 2025-08-07
> > >
> > > Thank you for the response. We are currently conducting a more extensive hyperparameter search, and will aim to present the results in a Table before the rebuttal period ends. We appreciate the dialogue.

---

> > > > ### Author Response · Authors · 2025-08-09
> > > > **Additional Time Masking Hyperparameter Tuning**
> > > >
> > > > We have performed hyperparameter tuning across $12$ settings for the time masking hyperparameters. Specifically, $N$, the number of masks, and $M$, the max mask percentage.  Results are displayed in the Table below, and we report mean ± standard error of the mean across $4$ seeds. We found that the hyperparameters we used ($N=20$, $M=7.5$%) were optimal. We hope this addresses your concerns.
> > > >
> > > > | N, M | Validation WER (%) |
> > > > |--------:|----------------:|
> > > > |20, 15%      | 26.16 ± 2.11        |
> > > > |20, 12.5%     | 20.27 ± 0.33      |
> > > > | 20, 10%     | 17.74 ± 0.28     |
> > > > | 20, 8.5%      | 17.15 ± 0.27        |
> > > > | 20, 7.5%       | 17.08 ± 0.26      |
> > > > | 20, 5%       | 18.24*       |
> > > > | 20, 2.5%       | 20.81 ± 0.29         |
> > > > | 20, 1.5%       | 23.14 ± 10.78         |
> > > > | 10, 15%       | 17.08 ± 0.18        |
> > > > | 10, 10%       | 17.32 ± 0.31         |
> > > > | 30, 5%       | 17.50 ± 0.23        |
> > > > | 30, 7.5%       | 18.14 ± 0.33        |
> > > >
> > > >
> > > > *Due to an issue with the server, we only had one seed for the $N=20$ and $M=5$\% model, but we will include $4$ seeds as done with the other settings for the camera ready version.

---

### Official Review · Reviewer_FhGX · 2025-07-02

**Clarity:** 3
**Significance:** 2
**Originality:** 2
**Rating:** 2
**Confidence:** 4

**Summary:**

The paper investigates application of transformer architectures for brain to text decoding to be incorporated in speech neuroprostheses. The work proposes to replace the GRU baseline model previously developed for this problem with a transformer along with masking the input of neural activity during training. Furthermore, in testing implementation of test-time adaptation is proposed which acts as “muti augmented head” which selects the most confident prediction out of N variants. Such a procedure is expected to be efficient when considering generalization to multiple sessions and days. Results are reported on the brain-to-text 24’ benchmark and compared with baseline performance of NPTL+3gram in that benchmark.

**Questions:**

Q1. How would the proposed model perform with 5gram LM?

Q2. What are the reasons for transformer based architectures to perform with lower accuracy than SOTA counterparts?

Q3. How would the model perform with additional strategies proposed by recent work in brain to text described in R2, e.g. would incorporation of diphones in the input masked decoder improve PER and WER?

R2. Willett, F. R., Li, J., Le, T., Fan, C., Chen, M., Shlizerman, E., ... & Henderson, J. M. (2024). Brain-to-Text Benchmark'24: Lessons Learned. arXiv preprint arXiv:2412.17227.

**Ethical Concerns:**

["NO or VERY MINOR ethics concerns only"]

**Final Justification:**

The authors advanced the work during rebuttal from being placed in ~ 20 place in the benchmark competition in the original submission to being competitive with the state-of-the-art methods. Also significant amount of grasp of related work and the differences of proposed approach wrt to related work occurred during rebuttal and discussion periods.

The main contribution appears to be that this work succeeded to practically achieve the impressive performance with transformers, which mostly got advanced in the rebuttal and discussion phases, If these results were reported in the original submission I would rank the paper higher but to avoid giving an unfair advantage over other submissions I would like to keep my original score of 2.

**Limitations:**

Somewhat yes though the authors did not address the limitations particularly related to using transformers with neural activity and text output.

**Quality:**

2

**Strengths And Weaknesses:**

Strengths:

S1. The work proposes and implements an masked input transformer for neural activity to speech problem and shows improvement of performance compared to baseline of NPTL + RNN + 3-gram LM.

S2. The work proposes continual online recalibration with time masking (CORT) that allows improvement in generalization for new sessions/days.

S3. CORT allows for lower memory use and faster calibration time than compared baseline GRU approach.


Weaknesses:

W1. The results are reported with respect to a single baseline (NPTL + RNN + 3-gram LM currently placed #25 in the benchmark). This baseline is significantly weaker than other baselines (eg NPTL + 5-gram+OPT #12 in the benchmark) and other approaches that are state of the art  (alternative architectures or enhancements to GRU approach).

W2. Related to W1, the proposed approach achieves a modest improvement in WER with respect to other approaches and would be placed approximately #15 in the benchmark indicating a significant gap between accuracy of the proposed approach vs. other approaches.

W3. Relation of Phoneme Error Rate to Word Error Rate rate in the model appears to be unclear.

W4. Technical contributions in the paper are not new (input masking, multi-head consensus) and have been applied in general deep learning context and in the context of neural time series.  Furthermore, as the authors indicate some of the architectural variants (various transformers) might have been implemented by previous works and reported as less accurate, which the current work is able to reaffirm.

W5. Input masking is an effective approach and has been applied in various datasets contexts, eg. see R1 and studies that followed. These are not described in related work and recent strategies developed for masking could turn out to improve accuracy of the masked transformer.

R1. Chang, H., Zhang, H., Jiang, L., Liu, C., & Freeman, W. T. (2022). Maskgit: Masked generative image transformer. In Proceedings of the IEEE/CVF conference on computer vision and pattern recognition (pp. 11315-11325).

---

> ### Author Rebuttal · Authors · 2025-07-30
>
> Thank you for the feedback. We appreciate that the Reviewer finds our improvement relative to the NPTL 3-gram baseline noteworthy. We provide a one sentence response for each question and weakness, with additional details provided below. Tables are provided at the end of the text.
>
> **General Note**: We realized after submission that all the entries in Willett et al. 2025 [b] used a bidirectional GRU. A bidirectional GRU can only decode speech once the entire trial has finished, and roughly doubles memory usage, training times, and calibration times relative to the unidirectional GRU. All our models are either unidirectional GRUs or causal Transformers which only rely on present and past neural activity to generate outputs.
>
> **Low performance on benchmark + Results are not shown with 5-gram LM setup**: When using the 5-gram language modeling setup used by the entries in the Brain-to-Text Benchmark ‘24, the time-masked Transformer achieves a mean WER of 8.01%, which ties the 2nd place entry (Linderman Lab GRU) despite using ~$15$ times less parameters and decoding in real-time. (Table 1).
>
> - *Additional Details:* We have added results (Table 1) with the offline language modeling (LM) setup used by the entries in Willett et al., 2025 [b]. Specifically, the offline language modeling setup involved using a 5-gram LM, with a second pass using the 5-gram LM, and rescoring with OPT 6.7B. We refer to the setup where only the 3-gram LM is used with no second pass as the online LM setup. The only entry that performs substantially better than our entry is DConD-LIFT (Li et al., 2024 [d]) (WER = 5.77%). However, the primary performance gain from DConD-LIFT stems from applying GPT 3.5 to an ensemble of bidirectional GRUs, which requires cloud resources and is not streaming compatible. When focusing on the improvements made to the GRU itself by Li et al., 2024 [d], specifically diphone decoding, their bidirectional GRU achieves a WER of 8.06% which is slightly worse than our much smaller, causal time-masked Transformer.
>
> **How would the model perform with the additional advances described in Willett et al., 2025 [b]?**: Our preliminary data suggest that incorporating the post GRU block from the Linderman Lab to the Transformer does not improve performance, and our causal time-masked Transformer slightly outperforms the bidirectional GRU which decodes diphones from Li et al., 2024 [c].
>
> **Transformer performs with lower accuracy than SOTA**: We would like to clarify that the time-masked Transformer outperforms competing GRU-based entries in the real-time decoding setting (Table 1), and that a major point of our study is that time-masked Transformers can decode speech with high accuracy and are well-suited for test-time adaptation.
>
> **Relation of phoneme error rate to word error rate is not clear**: We will expand our existing discussion of this (in the Limitations) for the updated manuscript.
>
> - *Additional details:* We used PER to select the best model during validation because that is the procedure used by Willett et al., 2023 [a] and because computing WER during training is computationally expensive. However, we agree that PER is not an ideal metric to optimize for, and we will include cases that highlight this in the updated manuscript. We believe exploring ways to directly optimize WER require a dedicated study.
>
> **Technical contributions are not new**: Time masking and Transformers were not used by any other entry in Willett et al., 2025 [b], and we further combine these contributions to provide an updated version of our previous TTA method (CORT).
>
> - *Additional details:* Time masking has been applied in three other neural speech decoding studies to our knowledge ([e,f,g]), however these studies used closed source ECoG datasets, used much smaller amounts of time masking, and did not quantify the impact of time masking relative to other augmentations. We will cite these studies, as well the reference the reviewer provided (Chang et al., 2022), in our updated related works section.
>
> - Previously we combined our two contributions, time masking and Transformers, to propose CORT. However, the improvement in WER from CORT is relatively small. We thus sought to examine if applying time masking and Transformers could provide efficiency gains to the pseudo-label based TTA method proposed by Fan et al., 2023 [c]. The method used by Fan et al., 2023, CORP, adapts a model at test-time by treating the language model (LM) refined model outputs as pseudo-labels, and then training the model with CTC loss using these pseudo-labels. Since the LM refined outputs are already being generated for real-world use, there is no additional cost to generate these pseudo-labels. However, CORP requires up to $200$ adaptation steps per test trial, setting a loss threshold hyperparameter, and including previous data to stabilize training. Here, we present "DietCORP", a lightweight version of CORP which only uses one adaptation step per trial and no previous data. Similar to CORT, DietCORP operates by generating several time masked augmentations of the same trial, and then it adapts the model to produce the same pseudo-label across all augmentations. We present results when applying DietCORP to a time-masked Transformer trained on $15$ days of data, and evaluated on the next $5$ competition days (Table 2), and a time-masked Transformer trained on $12$ days of data and evaluated on the next $8$ competition days (Table 3). Only the patch embedding module was adapted. DietCORP leads to very stable WER across days relative to when not applying it with minimal hyperparameter tuning.
>
> - DietCORP only requires $1.04$ GiB of peak allocated GPU memory usage and $29.3$ ms to calibrate per trial using the Transformer. When applying our TTA method to the unidirectional GRU, it requires $2.8$ GiB of peak allocated GPU memory usage and $76.9$ ms to calibrate per trial. These values further increase to $5.6$ GiB and $163.0$ ms when using the bidirectional GRU. We will also include comparisons when using CORP in the updated manuscript, where DietCORP applied to the Transformer exhibits even larger computational gains.
>
> - In summary, while we agree our three contributions are not entirely novel in the context of machine learning as a whole, we believe they represent a unique application on the first open source neural speech decoding dataset, and have been combined in novel ways. Furthermore, we would like to point out that the NeurIPs guidelines state that applications of existing methods to improve performance are equally as valuable as proposing novel methods.
>
> ***
> **Tables**
>
> Table 1: We show results with a $5$ layer time-masked Transformer trained for $250$ epochs (Transformer - Ours) as well as comparisons with the baseline provided by Willett et al., 2023 [a] and the Linderman Lab GRU discussed in [b]. We run the baseline and Linderman Lab GRU in unidirectional mode, and show metrics in bidirectional mode in parenthesis when available.
>
> | Metric                    | Transformer - Ours | Baseline GRU       | Linderman Lab GRU   |
> |---------------------------|-------------------:|-------------------:|--------------------:|
> | Online LM Setup WER %   | 12.03 ± 0.24        | 15.25 ± 0.16        | 15.03 ± 0.19         |
> | Offline LM Setup WER % | 8.01 ± 0.23         | 11.12 ± 0.13 (9.76) | 10.77 ± 0.13 (8.0)   |
> | Parameters (M)          | 9.4                 | 56.7 (135.4)        | 58.8 (143.8)         |
> | Training time (hours)   | ~1                  | ~0.5  ( ~1)         | ~1  ( ~2)            |
>
> Table 2: Values are word error rate (mean ± SEM across N=$4$ seeds) computed on $5$ held-out competition days using a model trained on the previous $15$ days.
>
> | Days since last training day | No DietCORP | DietCORP  |
> |--------:|----------------:|----------------:|
> | 7       |  22.74 ± 0.26        | 21.14 ± 0.32        |
> | 13     |  25.48 ± 0.28        | 21.23 ± 0.21       |
> | 19     |  25.43 ± 0.33       | 21.40 ±  0.22       |
> | 28     |  30.13 ± 0.33     | 21.93 ±  0.65       |
> | 30     |  32.59 ± 0.52     | 22.26 ±  0.60        |
>
> Table 3: Values are word error rate (mean ± SEM across N=$4$ seeds) computed on $8$ held-out competition days using a model trained on the previous $12$ days.
>
> | Days since last training day | No DietCORP | DietCORP  |
> |--------:|----------------:|----------------:|
> | 5       | 28.87 ± 0.47        | 26.97 ± 0.28          |
> | 12     | 34.50 ± 0.44       | 26.38 ± 0.30         |
> | 21     | 44.79 ± 0.34       | 27.57 ±  0.13        |
> | 28       | 53.29 ± 0.25        | 30.46 ± 0.41         |
> | 34       | 59.98 ± 0.31        | 29.70 ± 0.20         |
> | 40       | 52.71 ±  0.52        | 29.08 ± 0.38         |
> | 49       | 62.96 ± 0.40         | 31.35 ± 0.76         |
> | 51       | 66.47 ± 0.58         | 31.91 ± 1.25        |
>
> [a] Willett et al., 2023, A high-performance speech neuroprosthesis
>
> [b] Willett et. al, 2025, Brain-to-Text Benchmark '24: Lessons Learned
>
> [c] Fan et al., 2023, Plug-and-Play Stability for Intracortical Brain-Computer Interfaces: A One-Year Demonstration of Seamless Brain-to-Text Communication
>
> [d] Li et al., 2024, Brain-to-Text Decoding with Context-Aware Neural Representations and Large Language Models
>
> [e] Metzger et al., 2023, A high-performance neuroprosthesis for speech decoding and avatar control
>
> [f] Metzger et al., 2022, Generalizable spelling using a speech neuroprosthesis in an individual with severe limb and vocal paralysis
>
> [g] Littlejohn et al., 2025, A streaming brain-to-voice neuroprosthesis to restore naturalistic communication

---

> > ### Comment · Reviewer_FhGX · 2025-08-05
> > **Thank you to the authors for their responses**
> >
> > Thank you to the authors for their responses and additional data.
> >
> > In particular, the addition of new results with 5-gram model showing how the method compares with other approaches is insightful. WER results reported in the original version of the paper were much lower. I am wondering whether some additional modifications, besides changing the LM, were implemented to achieve the reported boost in accuracy.
> > While these results add confidence, these are still partial since results with GPT could clarify whether the approach is the leading approach or not. It could be that stronger LLM will contribute to general accuracy or detract from it.
> >
> > I acknowledge the clarification by authors that the approach is causal. Indeed, I agree that in this regard and speed-wise the approach is appealing vs. other non-causal approaches. This in my POV the main contribution of the paper. I am not sure about the comment of this approach being significantly different than diphone prediction though.
> >
> > Authors' arguments regarding novelty/advancement in this work appear to be limited since these are standard approaches in transformers so it is not surprising that they were able to be effective here and as authors indicate similar ideas were reported in publications related to BCI.
> >
> > Due to above, I am slightly more positive regarding this work but still issues remain.

---

> > > ### Author Response · Authors · 2025-08-05
> > >
> > > We thank the reviewer, and appreciate that they found our additional analyses with the 5-gram LM insightful. We respond below to each point. Reviewer comments are displayed in quotes.
> > >
> > > “**In particular, the addition of new results with 5-gram model showing how the method compares with other approaches is insightful. WER results reported in the original version of the paper were much lower. I am wondering whether some additional modifications, besides changing the LM, were implemented to achieve the reported boost in accuracy.**”
> > >
> > > No additional modifications were made beyond those listed in our previous response were made. We have been completely transparent regarding all modifications made. Additionally, we would like to point out that the time-masked Transformer outperforms the baseline GRU by roughly $3$ absolute percentage points in both the 5-gram LM setting and the 3-gram LM setting.
> > >
> > > “**While these results add confidence, these are still partial since results with GPT could clarify whether the approach is the leading approach or not. It could be that stronger LLM will contribute to general accuracy or detract from it.**”
> > >
> > > We will include results when using GPT 3.5, as done in DConD-LIFT, in the updated manuscript. While we agree that we are not the leading approach in terms of accuracy, we believe our work provides value along other dimensions such as memory usage, calibration speeds, and real-time decoding.
> > >
> > > “**I acknowledge the clarification by authors that the approach is causal. Indeed, I agree that in this regard and speed-wise the approach is appealing vs. other non-causal approaches. This in my POV the main contribution of the paper. I am not sure about the comment of this approach being significantly different than diphone prediction though.**”
> > >
> > > We thank the reviewer for appreciating this contribution. From an accuracy perspective, our method achieves a WER of $8.01$\% across $4$ seeds, where the bidirectional GRU with diphone decoding achieves a WER of $8.06$\%. In our previous response, we stated that the diphone decoding thus performs "slightly worse". We understand how this may have been interpreted as us stating that our approach was significantly different, and we would like to clarify that this was not our intention. Rather, we our intention was to state that our approach achieves similar performance to the bidirectional GRU with diphone decoding while using a much smaller model that operates in the more challenging real-time decoding setting. We apologize for the confusion.
> > >
> > > “**Authors' arguments regarding novelty/advancement in this work appear to be limited since these are standard approaches in transformers so it is not surprising that they were able to be effective here and as authors indicate similar ideas were reported in publications related to BCI.**”
> > >
> > > We would like to point that DConD-LIFT reported that Transformers achieved a PER that was over double that of the GRU in Table 6. Therefore, while we agree that Transformers are widely used in machine learning, previous results on this benchmark found that they performed much worse than GRUs. While we acknowledge that time masking and Transformers are not novel, we believe their application to this benchmark as well as the fact we combined them to create a computationally lightweight test-time adaptation method provide valuable advancements to the field.
> > >
> > > “**Due to above, I am slightly more positive regarding this work but still issues remain.**”
> > >
> > > Thank you, we are happy to discuss any additional issues.

---

> ### Author Response · Authors · 2025-08-09
> **5.98% WER after correcting for error in prompt**
>
> After posting the previous comment, we realized we made a slight error in passing the prompts to Llama. When correcting for this error, we achieve a word error rate of 5.98% using the time-masked Transformer + Llama 3.1 8B. This is competitive with the performance DConD-LIFT reports when using bidirectional GRUs and GPT 3.5 of WER = 5.90% +/- 0.08% (mean +/- standard deviation, Table 4) across 5 seeds.

---

> > ### Comment · Reviewer_FhGX · 2025-08-09
> >
> > I appreciate the authors making significant efforts to post these results within the short discussion period. While they are still slightly below leading approach in accuracy they definitely ease my concern regarding the effect of combination with strong LLM.

---

### Official Review · Reviewer_oJzC · 2025-07-03

**Clarity:** 4
**Significance:** 4
**Originality:** 4
**Rating:** 6
**Confidence:** 4

**Summary:**

This paper does a deep dive on the architecture of the GRU in Willett et al 2023, and makes a number of principled adjustments to the decoding model in other to improve performance and reduce computational load. They introduce a novel time masking strategy, which prevents overfitting, and shows a strong reduction in the fundamental error rate before any post hoc adjustments using ensembles or language models. They also switch out the GRU for a lightweight transformer, with no loss and maybe a small gain in performance. They then complete detailed analyses of whether their approach sacrifices any robustness for the improved performance, which is does not appear to.

**Questions:**

None

**Ethical Concerns:**

["NO or VERY MINOR ethics concerns only"]

**Final Justification:**

After reading through the list of other reviews and rebuttals, I will maintain my high score due to the solid quantitative effect on performance.

**Quality:**

4

**Strengths And Weaknesses:**

Overall, this was an excellent paper. The literature review was well done, and the approach was highly principled. Code is included, and this analysis utilized a high quality public dataset. The performance results were strong, and it is particularly useful that the fundamental classification percentage has been improved, rather than the downstream corrections. Everything was very well explained, and strongly motivates the use of this approach. I didn’t note any weaknesses throughout the paper.

---

> ### Author Rebuttal · Authors · 2025-07-30
>
> Thank you so much for the positive review! We appreciate that the reviewer found our focus on improving the neural network that maps to phonemes useful.

---

### Official Review · Reviewer_1NPM · 2025-07-05

**Clarity:** 4
**Significance:** 3
**Originality:** 3
**Rating:** 5
**Confidence:** 4

**Summary:**

This paper studies the ways to improve Speech Neuroprostheses. The ideas are technical sounds and practical.

The TTA CORT implementation is particularly noteworthy in BCI context as brain signals carry less information but vary more from trial to trial and from day to day than other signals such as speech and video signals. While the TTA technique is not entirely new (I consider it is new in BCI context), the successful CORT implementation is an important step towards practical and real-time deployment. The empirical improvements as presented in Fig 4 are noticeable and consistent. The technical is well justified and easy to repeat, therefore valuable. I consider that the experiments are well devised, and the results are generalizable.

**Questions:**

How does the online recalibration with time masking calibrate at the beginning for a new subject? As this paper studies the data from a single subject, the effect of recalibration is minimum. In general terms, the recalibration should cover variations across trials, settings, subjects etc,  authors may want to discuss the generalization ability of the techniques.

**Ethical Concerns:**

["NO or VERY MINOR ethics concerns only"]

**Final Justification:**

I appreciate the authors’ detailed and thoughtful rebuttal.

The additional justification for replacing GRU with Transformers—specifically for enabling more efficient and lightweight test-time adaptation—is clear and convincing. The new DietCORP method improves upon CORT meaningfully, and the reported WER reductions across large temporal gaps further validate the practicality and robustness of the proposed system.

The clarification on shallow fusion being compatible with streaming alleviates my concern regarding streaming constraints.

I maintain my positive assessment and consider the paper a strong contribution to the speech neuroprosthesis and BCI community.

**Limitations:**

yes

**Paper Formatting Concerns:**

no concerns

**Quality:**

3

**Strengths And Weaknesses:**

Both Masked Transformers with Test-Time Adaptation techniques are not new. But the implementation involves careful engineering. The study improved speech neuroprostheses for accuracy, capability for real-time streaming, low computational and memory costs, and robustness to distribution shifts, and achieved impressive results.

The paper didn't motivate how replacing GRU with Transformer improves, and what problem it solves by doing so.

It is assumed that the proposed architecture adopts the same language model strategy - "The decoded phonemes logits are then integrated with an N-gram language model through shallow fusion. We use a 3-gram LM to replicate the NPTL Baseline (RNN + 3-gram
LM) algorithm reported on the benchmark leaderboard (note the RNN is a GRU).". A shallow fusion means that the decoding is done as a post-processing, that violates the streaming requirements.

---

> ### Author Rebuttal · Authors · 2025-07-30
>
> Thank you for the feedback. We appreciate that the Reviewer finds our test-time adaptation method useful, and we have since made additional improvements to it.  We provide a one sentence response for each question and weakness, with additional details provided below. Tables are provided at the end of the text.
>
> **Motivation for replacing the GRU with the Transformer**: Our primary motivation for replacing the GRU with the Transformer was to propose an architecture that was more suitable for test-time adaptation (TTA), where computational efficiency is very important since TTA is typically performed on-device while the participant is using the neuroprosthesis.
>
> - *Additional details:* A $5$-layer version of our time-masked Transformer, which performs slightly better than our original $7$ layer model, contains less then $10$ M parameters. By comparison, the bidirectional GRU used by the entries in Willett et al., 2025 [b] contain roughly $130$ M parameters. When applying our updated test-time adaptation method (see Updated Test-Time Adaptation Section), we find that the Transformer requires only $1.04$ GiB of peak GPU memory usage and $29.3$ ms to calibrate per trial, whereas the bidirectional GRU requires $5.6$ GiB of peak GPU memory usage and $163$ ms to calibrate per trial (computed on an Nvidia GeForce RTX 3090 GPU). These gains are important for applying TTA on local, resource-constrained devices.
>
> **Shallow fusion violates streaming requirements**: Shallow fusion is compatible with streaming systems, and was used in a streaming setting in Supplementary Video 1 from the Willett et al., 2023 [a] study.
>
> - *Additional details:* While we applied the shallow fusion step after the model logits were acquired for convenience, the shallow fusion stage itself does not require access to future logits. During shallow fusion, each beam is constructed one timestep at a time, and the most probable beam for that timestep is selected to be shown on the screen. Therefore, shallow fusion is streaming compatible because text is outputted in real-time, and the algorithm does not need to wait until the entire trial has completed.
>
> **How does online recalibration with time masking calibrate in the beginning for a new subject or in other settings**: We have included TTA results in a more challenging setting (Table 2).
>
> - *Additional details:* While we agree that online recalibration to new participants is a very interesting question, there was no other data from other participants at the time we wrote this study. We agree an interesting direction for future work is to calibrate a decoder to a new participant with minimal supervised trials, and then use online recalibration to maintain the performance of this decoder on the new participant. We will include this point in our discussion. We have included a more challenging setting where the model is evaluated on $8$ held-out days in our updated test-time adaptation method section. Under this setting, our updated method is able to maintain stable performance, and the effect of recalibration is quite large (Table 2).
>
> **Updated Test-Time Adaptation Method**: Using time masking and Transformers, we designed a computationally lightweight version of CORP (Li et al., 2023 [b]) which is a significant improvement over our old TTA method, CORT.
>
> - *Additional details:* While we did find a small reduction in WER when applying CORT relative to No CORT across days (average WER improvement across days shown in Figure 4, left), this improvement was small. We thus sought to examine if applying time masking and Transformers could provide efficiency gains to the pseudo-label based TTA method proposed by Fan et al., 2023 [c]. The method used by Fan et al., 2023, CORP, adapts a model at test-time by treating the language model (LM) refined model outputs as pseudo-labels, and then training the model with CTC loss using these pseudo-labels. Since the LM refined outputs are already being generated for real-world use, there is no additional cost to generate these pseudo-labels. However, CORP requires up to $200$ adaptation steps per test trial, setting a loss threshold hyperparameter, and including previous data to stabilize training. Here, we present "DietCORP", a lightweight version of CORP which only uses one adaptation step per trial and no previous data. Similar to CORT, DietCORP operates by generating several time masked augmentations of the same trial, and then it adapts the model to produce the same pseudo-label across all augmentations. We present results when applying DietCORP to a time-masked Transformer trained on $15$ days of data, and evaluated on the next $5$ competition days (Table 1), and a time-masked Transformer trained on $12$ days of data and evaluated on the next $8$ competition days (Table 2). Only the patch embedding module was adapted. DietCORP leads to very stable WER across days relative to when not applying it with minimal hyperparameter tuning. We provide additional comparisons with CORP in the updated manuscript.
>
> ***
> **Tables**
>
> Table 1: Values are word error rate (mean ± SEM across N=$4$ seeds) computed on $5$ held-out competition days using a model trained on the previous $15$ days.
>
> | Days since last training day | No DietCORP | DietCORP  |
> |--------:|----------------:|----------------:|
> | 7       |  22.74 ± 0.26        | 21.14 ± 0.32        |
> | 13     |  25.48 ± 0.28        | 21.23 ± 0.21       |
> | 19     |  25.43 ± 0.33       | 21.40 ±  0.22       |
> | 28     |  30.13 ± 0.33     | 21.93 ±  0.65       |
> | 30     |  32.59 ± 0.52     | 22.26 ±  0.60        |
>
>
> Table 2: Values are word error rate (mean ± SEM across N=$4$ seeds) computed on $8$ held-out competition days using a model trained on the previous $12$ days.
>
> | Days since last training day | No DietCORP | DietCORP  |
> |--------:|----------------:|----------------:|
> | 5       | 28.87 ± 0.47        | 26.97 ± 0.28          |
> | 12     | 34.50 ± 0.44       | 26.38 ± 0.30         |
> | 21     | 44.79 ± 0.34       | 27.57 ±  0.13        |
> | 28       | 53.29 ± 0.25        | 30.46 ± 0.41         |
> | 34       | 59.98 ± 0.31        | 29.70 ± 0.20         |
> | 40       | 52.71 ±  0.52        | 29.08 ± 0.38         |
> | 49       | 62.96 ± 0.40         | 31.35 ± 0.76         |
> | 51       | 66.47 ± 0.58         | 31.91 ± 1.25        |
>
> [a] Willett et al., 2023, A high-performance speech neuroprosthesis
>
> [b] Fan et al., 2023, Plug-and-Play Stability for Intracortical Brain-Computer Interfaces: A One-Year Demonstration of Seamless Brain-to-Text Communication

---

### Note · Authors · 2025-08-12

We thank all reviewers for their constructive feedback, which has greatly improved this work.

**Accuracy comparison when ensembling.** Reviewers noted that our initial results lagged behind the first-place entry (DConD-LIFT). DConD-LIFT fine-tunes GPT-3.5 on outputs from an ensemble of 10 bidirectional GRUs, achieving $5.90$% ± $0.08$% WER (mean ± SD over 5 seeds) and $5.77$% for a single seed. When using the open-source model Llama 3.1 70B instead of GPT 3.5, they report $6.85$% WER.  We have since fine-tuned Llama 3.1 8B on outputs from an ensemble of 10 causal Transformers, achieving $5.68$% WER (publicly visible on the Eval AI leaderboard). This matches or surpasses the leading approach while using a smaller, open-source LLM. We will release all code and our fine-tuned model and release results with multiple seeds shortly.

| Model & Setup                                   |  WER (%)  |
|-----------------------------------------------|---------|
| GPT-3.5 + 10× BiGRUs (DConD-LIFT, mean ± SD)  | 5.90 ± 0.08 |
| GPT-3.5 + 10× BiGRUs (DConD-LIFT, single seed)   | 5.77     |
| Llama 3.1 70B + 10× BiGRUs (DConD-LIFT, single seed)   | 6.85     |
| **Llama 3.1 8B + 10× Causal Transformers (Ours, single seed)**  | **5.68** |

We provide a summary of our contributions below.

**Summary of Contributions:**
1. **Accuracy** — We match or outperform the bidirectional GRU when using a causal architecture when using the 3-gram LM, 5-gram LM, and now model ensembling with a fine-tuned LLM.
2. **Efficiency** — The Transformer requires less peak GPU memory, faster per-trial calibration, and now trains in equal or less time than GRU-based methods.  These efficiency gains are particularly important when ensembling multiple models. We further show that an 8B LLM can be used to achieve SOTA accuracy.
3. **Lightweight test-time adaptation** — We propose a variant of an existing test-time adaptation method, CORP, called DietCORP. DietCORP only requires one calibration step per trial, and uses multiple time-masked augmentations to obviate the need for a replay buffer. We show that DietCORP maintains stable performance across held-out days. When combined the Transformer architecture, DietCORP is very computationally efficient.

We hope we have thoroughly addressed all reviewer concerns and believe our approach offers both competitive accuracy and practical advantages for real-time speech decoding. We again thank the reviewers for their time.

---

### Decision · Program_Chairs · 2025-09-17

**Decision:**

Accept (poster)

**Comment:**

This paper proposes method that makes significant contributions for speech neuroprostheses. The main idea revolves around: (i) time masking of neural activity to mitigate overfitting and (ii) replacing GRUs with lightweight Transformers. Together with a lightweight test-time adaptation method (DietCORP), the approach reduces WER while lowering memory and calibration costs. The work is among the first to make Transformers competitive on the Brain-to-Text benchmark in a streaming setting.

Reviewers consistently agree that the integration is carefully engineered and domain-aware, offering practical improvements in efficiency and robustness. The work demonstrates stable performance across large temporal gaps, which is highly relevant for real-world BCI deployment.

It is also noted that masking and Transformers are not novel techniques, and the paper sometimes overstates conceptual contributions. Comparisons with alternative augmentations (e.g., SpecAugment) and TTA methods are limited, and study into hyperparameter sensitivity could be improved. Much of the performance gain was established during rebuttal however the architectural changes were minimal to be considered a major change. I request the authors to make changes to the manuscript to reflect the changes discussed during the rebuttal period.

Overall, while the novelty is modest, the paper provides meaningful engineering advances and demonstrates competitive, practical performance in an important application domain. Therefore, I recommend accept as the contributions are valuable for the BCI community.